# Molecular basis of the TRAP complex function in ER protein biogenesis

Mateusz Jaskolowski ®[1,4], Ahmad Jomaa ®[1,2,4] ✉, Martin Gamerdinger ®[3,4], Sandeep Shrestha[3], Marc Leibundgut[1], Elke Deuerling ®[3] ✉ & Nenad Ban ®[1] ✉

The translocon-associated protein (TRAP) complex resides in the endoplasmic reticulum (ER) membrane and interacts with the Sec translocon and the ribosome to facilitate biogenesis of secretory and membrane proteins. TRAP plays a key role in the secretion of many hormones, including insulin. Here we reveal the molecular architecture of the mammalian TRAP complex and how it engages the translating ribosome associated with Sec61 translocon on the ER membrane. The TRAP complex is anchored to the ribosome via a long tether and its position is further stabilized by a finger-like loop. This positions a cradle-like lumenal domain of TRAP below the translocon for interactions with translocated nascent chains. Our structure-guided TRAP mutations in *Caenorhabditis elegans* lead to growth deficits associated with increased ER stress and defects in protein hormone secretion. These findings elucidate the molecular basis of the TRAP complex in the biogenesis and translocation of proteins at the ER.

Secretory and membrane proteins constitute one-third of the human cell proteome[1] and are mostly cotranslationally targeted to the endoplasmic reticulum (ER) to be translocated into the ER lumen or inserted into the ER membrane[2,3]. The membrane protein complex Sec61 acts as a protein-conducting pore to facilitate translocation of such proteins[4]. Referred to as Sec translocon, Sec61 is a hetero-trimeric complex composed of α-, β- and γ-subunits, shaped like an hourglass with a central pore plugged by a hydrophobic helix that is displaced during protein translocation across the membrane, and a lateral gate where the two halves of the protein open to allow insertion of the newly synthesized polypeptide into the membrane[2,5].

To accommodate the diversity of proteins targeted to the ER, the Sec translocon interacts with different membrane protein complexes that aid in cotranslational folding, assembly and processing of protein clients[6]. One of the key interactors with the Sec translocon is a highly abundant heterotetrameric transmembrane translocon-associated protein (TRAP) complex (harboring subunits TRAPα, TRAPβ, TRAPγ and TRAPδ) that also binds translating ribosomes[7] and interacts with emerging nascent chains[8–11]. The TRAP complex is necessary for

secretion and translocation of a subset of proteins including hormones like angiotensin or atrial natriuretic peptide, and the insulin-like growth factor 1 receptor[10,12]. TRAP-dependent clients are characterized by a signal sequence with a weaker gating activity[10], with a lower hydro-phobicity and higher-than-average glycine and proline content in their signal sequence[12].

A recent study identified insulin as a bona fide client protein of TRAP, indicating that the TRAPα subunit is necessary for insulin biogenesis[13,14]. Previous analysis showed that the TRAP subunits become upregulated in pancreatic β cells when exposed to high concentration of glucose, presumably as a response to the requirement of insulin synthesis and secretion[15], and that a single-nucleotide mutation in the human TRAPα gene is linked with type 2 diabetes susceptibility[16]. Deletion of, or mutations in, TRAP subunits lead to impaired ER function and are linked to various disorders, which further underscores the importance of TRAP in cargo translocation across the membrane[17–20]. Recent biochemical work showed that TRAP inter-actions with nascent chains and the translating ribosome precede interactions with the Sec translocon. This presents the intriguing

[1]Department of Biology, Institute of Molecular Biology and Biophysics, ETH Zurich, Zurich, Switzerland. [2]Department of Molecular Physiology and Biological Physics and the Center for Cell and Membrane Physiology, University of Virginia, Charlottesville, VA, USA. [3]Department of Biology, Molecular Microbiology, University of Konstanz, Konstanz, Germany. [4]These authors contributed equally: Mateusz Jaskolowski, Ahmad Jomaa, Martin Gamerdinger. ✉e-mail: ahmadjomaa@virginia.edu; elke.deuerling@uni-konstanz.de; ban@mol.biol.ethz.ch

possibility that TRAP could also play a role in cargo handover from signal recognition particle (SRP) to the Sec translocon at the late stages of protein targeting[21].

Despite its important role in protein translocation across the ER membrane, the molecular basis of TRAP interaction with the ribosome and the Sec translocon are still not known. In this Article, we used single particle cryogenic electron microscopy (cryo-EM) combined with in vivo characterization to reveal the molecular basis of the interactions between the TRAP complex, the Sec translocon and a translating ribosome, which position TRAP for interactions with the nascent chain that emerges through the translocon pore. Our results describe a previously uncharacterized interaction between the C-terminal region of TRAPα with the ribosome. The structure also reveals the contact points between the TRAPα lumenal domain and the Sec translocon and shows the importance of hydrophobic residues within the TRAP lumenal domain, presumably important for folding of translocated proteins.

## Structure of the translating ribosome bound to TRAP-Sec61

We reconstituted a protein targeting reaction to the ER membrane by incubating programmed ribosomes displaying an ER signal peptide (ribosome nascent chain complex (RNC)) from rabbit reticulocyte lysate, together with canine SRP and pancreas EDTA and salt treated ER microsomes (EKRM)[22] (Extended Data Fig. 1). The sample was then solubilized using mild detergents, purified and investigated using cryo-EM. We anticipated the formation of RNC complexes engaged with the Sec translocon and various accessory proteins that facilitate cotranslational insertion and translocation of membrane and secretory proteins, respectively. Three-dimensional (3D) image classification resolved two complexes in the data: a ternary complex with RNC, TRAP and Sec61, and a second complex additionally including the oligosaccharyltransferase (OST) complex (Extended Data Fig. 2). Since the atomic structure of the OST complex was recently reported[23] and its interactions with Sec61 were described[24], the OST-containing complex is not further discussed here.

The 3D reconstruction of the ternary complex revealed the translating ribosome with a large detergent micelle at the exit of the ribosomal tunnel (Fig. 1a). The overall resolution of the reconstruction was 3.5 Å, whereas the region inside the micelle corresponding to the Sec61 was resolved between 3 Å and 7 Å (Extended Data Figs. 3 and 4). The EM density corresponding to the TRAP complex reveals its shape and orientation relative to the RNC as observed in low resolution cryo-EM and electron cryotomography (cryo-ET) maps[7,25–27]. The TRAP complex was resolved to around 6–8 Å resolution in the micelle area, showing a series of transmembrane alpha helices that allowed docking of a TRAPβγδ model predicted by AlphaFold2[28,29] (Fig. 1b). Density corresponding to a single transmembrane helix (TMH) of TRAPα, which, according to the AlphaFold2 model, stands separate from the other TRAP TMHs can also be observed (Extended Data Figs. 5a and 6a–c). On the side of the ER lumen, the TRAP complex was resolved to around 8–12 Å, presenting a cradle-like density that extended towards the exit of the Sec61 pore. The shape of this entire volume was well described by a trimeric, β-sheet-containing TRAPαβδ model predicted by AlphaFold2 in multimer mode (Fig. 1b). Additional confidence for correct docking came from the C terminus of the TRAPα subunit, which was resolved at ~3.5 Å and unambiguously revealed the interactions with the ribosome at near-atomic detail (Fig. 1c and Extended Data Figs. 3c and 4d). Its location also supports the placement of the free-standing TRAPα TMH, as a linker bridging these two elements is too short to reach the density corresponding to the other TRAP TMHs (Extended Data Fig. 6d,e). Furthermore, such placement of the TRAPα TMH positions a conserved region of positively charged residues next to the 5.8 S rRNA (Extended Data Fig. 6f,g), allowing for favorable electrostatic interactions. Inside the ribosomal exit tunnel, additional EM density representing the nascent chain is visible; however, it disappears at the exit (Extended

Data Fig. 7). Therefore, in spite of the heterogeneous local resolution of the RNC-bound TRAP complex, combining the direct interpretation of better resolved regions of TRAP with AlphaFold2 predictions[30] for the regions where secondary structure elements were visible allowed us to generate a complete model of the RNC:Sec61:TRAP complex (Fig. 1d and Extended Data Fig. 5).

The TRAP complex is positioned near Sec61, but none of the TRAP TMHs are in direct contact with the translocon. The main bundle of TMHs formed by three TRAP subunits (TRAPβ, TRAPγ and TRAPδ) is positioned next to the lateral gate of Sec61, whereas the single TMH of TRAPα is located on the opposite side (Fig. 1d). From the ER lumenal side, the immunoglobulin-like β-sandwich domains of TRAPα, TRAPβ and TRAPδ form a cradle-like domain that directly contacts the Sec translocon (Fig. 1d,e).

## TRAPα is anchored to the ribosome via its C-terminal tail

The prominent interaction between TRAP and the ribosome is mediated by the C-terminal tail of TRAPα folded in a hook-like shape that includes a short α-helix. These interactions occur in the vicinity of the exit of the ribosomal tunnel and involve insertion of a tryptophan between two helices of uL29 and electrostatic contacts with H9 of the 5.8S rRNA (Fig. 2a and Extended Data Fig. 4d). Considering the specificity of these interactions, it is likely that they contribute considerably to the affinity of the TRAP complex and that they are functionally important. Furthermore, alignment of TRAPα from different eukaryotic organisms, including plants and algae, shows a strong conservation of the anchor region including the tryptophan and a series of positively charged residues that point towards the negatively charged rRNA (Fig. 2b). Consequently, we refer to this region as the TRAPα anchor. The location of the anchor, despite being in close proximity, overlaps neither with the SRP nor the Sec61 binding sites on the ribosome[2,21] (Extended Data Fig. 8). There is also no overlap with the recently observed ribosome contact areas of the nascent polypeptide-associated complex (NAC)—an ER targeting regulator that recruits SRP to ribosomes[31] (Extended Data Fig. 8).

To validate the role of the anchor in TRAP function, we measured levels of a green fluorescent protein (GFP) reporter of ER stress driven by the hsp-4 promoter (hsp-4p::GFP) in *Caenorhabditis elegans*[32]. Consistent with a previous study[13], knockout (KO) of TRAPα in *C. elegans* resulted in a strong ER stress response, with elevated levels of the stress reporter hsp-4p::GFP detected throughout the worm body (Fig. 2c), indicating a general function of the TRAP complex in maintaining ER protein homeostasis in cells. We then designed a mutant of TRAPα that carries mutations in three residues (W228A, K235E and K239E, corresponding to human W255, N262 and K266, respectively) located within the anchor (Fig. 2a,b). Worms expressing this mutant also showed increased levels of the ER stress reporter hsp-4p::GFP relative to the wild-type, especially in highly secretory intestinal cells (Fig. 2d and Extended Data Fig. 9a), further highlighting the importance of this contact for TRAP function.

In addition to the TRAPα anchor, TRAP also interacts with the ribosome via less specific contacts between the negatively charged rRNA and a loop in the cytosolic domain of TRAPγ (referred to as the TRAPγ finger) harboring positively charged residues (Fig. 2e), which are conserved in different eukaryotic organisms, as revealed by the sequence alignment of this region of the protein (Fig. 2f). To better understand their importance, we designed a *C. elegans* mutant of TRAPγ that carries reverse charge mutations in three positively charged residues (R103E, K104E and K108E, corresponding to human R110, K111 and K115, respectively) (Fig. 2e,f). Again, RNAi-mediated knockdown of TRAPγ in *C. elegans* caused notable ER stress, as indicated by increased levels of the ER stress reporter hsp-4p::GFP (Fig. 2g). While expression of wt-TRAPγ from an RNAi-resistant transgene completely reversed the observed ER stress phenotype, the TRAPγ finger mutant provided only partial rescue (Fig. 2g and Extended Data Fig. 9b).

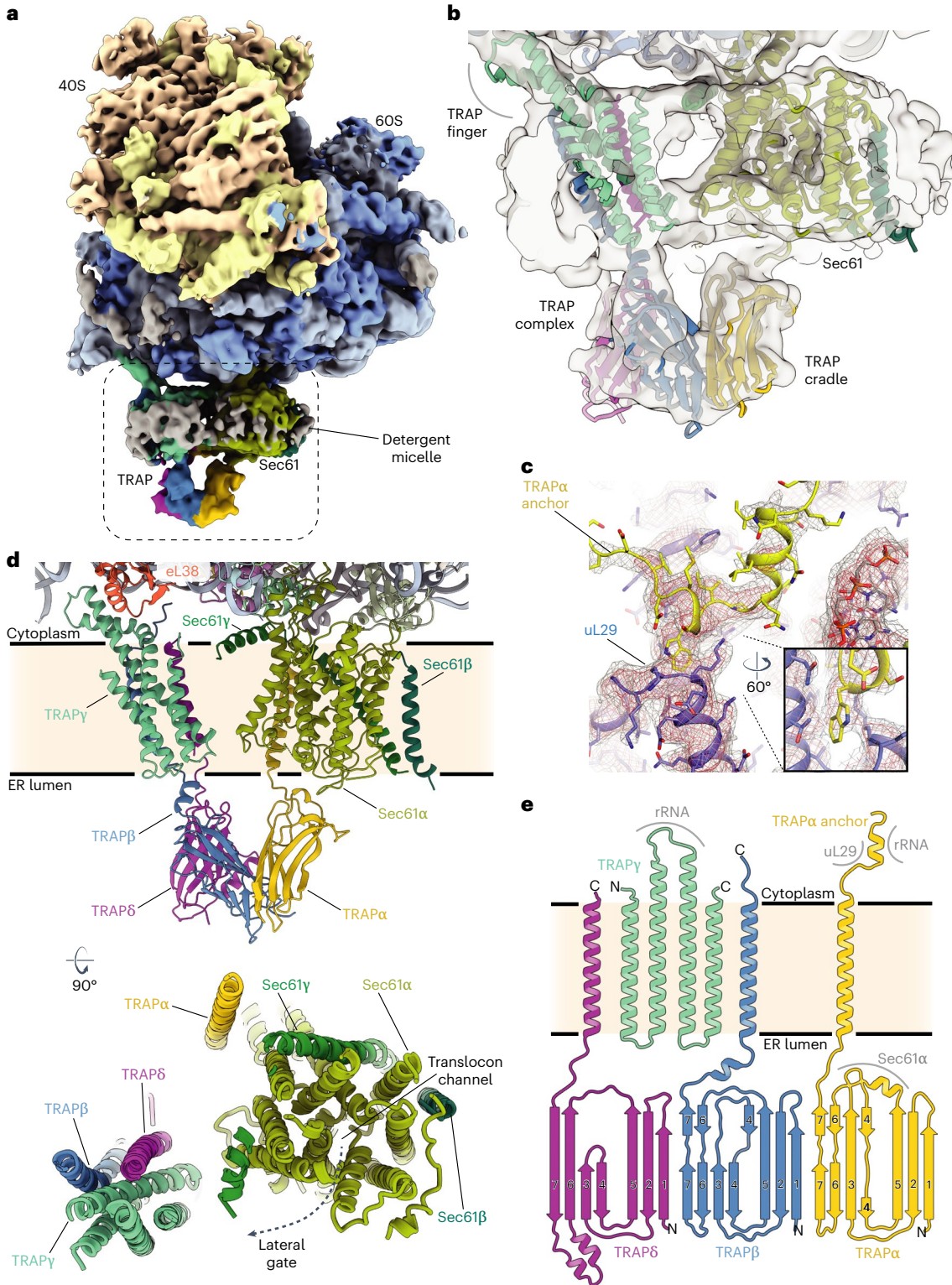

**Fig. 1 | Cryo-EM structure of the mammalian RNC, Sec translocon and TRAP ternary complex. a**, Cryo-EM map depicting TRAP, Sec translocon and the ribosome. Ribosomal proteins are colored brown and blue; the density of ribosomal RNA is colored yellow and light blue for the small and large subunit, respectively. The dashed box indicates the magnified region shown in **b**. **b**, Cross-section of the detergent micelle at the ribosomal exit tunnel. Densities and models of Sec translocon and TRAP complex are indicated. The map was filtered to 7 Å and shown at 3.2σ. **c**, Close-up of the C terminus of TRAPα (yellow) fitted into cryo-EM density shown as a mesh. Ribosomal protein uL29 (purple)

is indicated. Map shown at 3σ. **d**, Closeup of the RNC:Sec61:TRAP atomic model. Sec translocon and TRAP complex proteins are labeled and colored individually. Ribosomal protein eL38 is shown. An overview of the Sec61:TRAP complex in the membrane as observed from the cytosolic side is shown below. Proteins are colored and labeled individually. The pore of the Sec translocon channel and the lateral gate are indicated. **e**, Secondary structure diagram of the TRAP complex. TRAP proteins are colored as in previous panels. The N terminus and C terminus of each protein is labeled. Regions that interact with the ribosome and the Sec translocon are indicated.

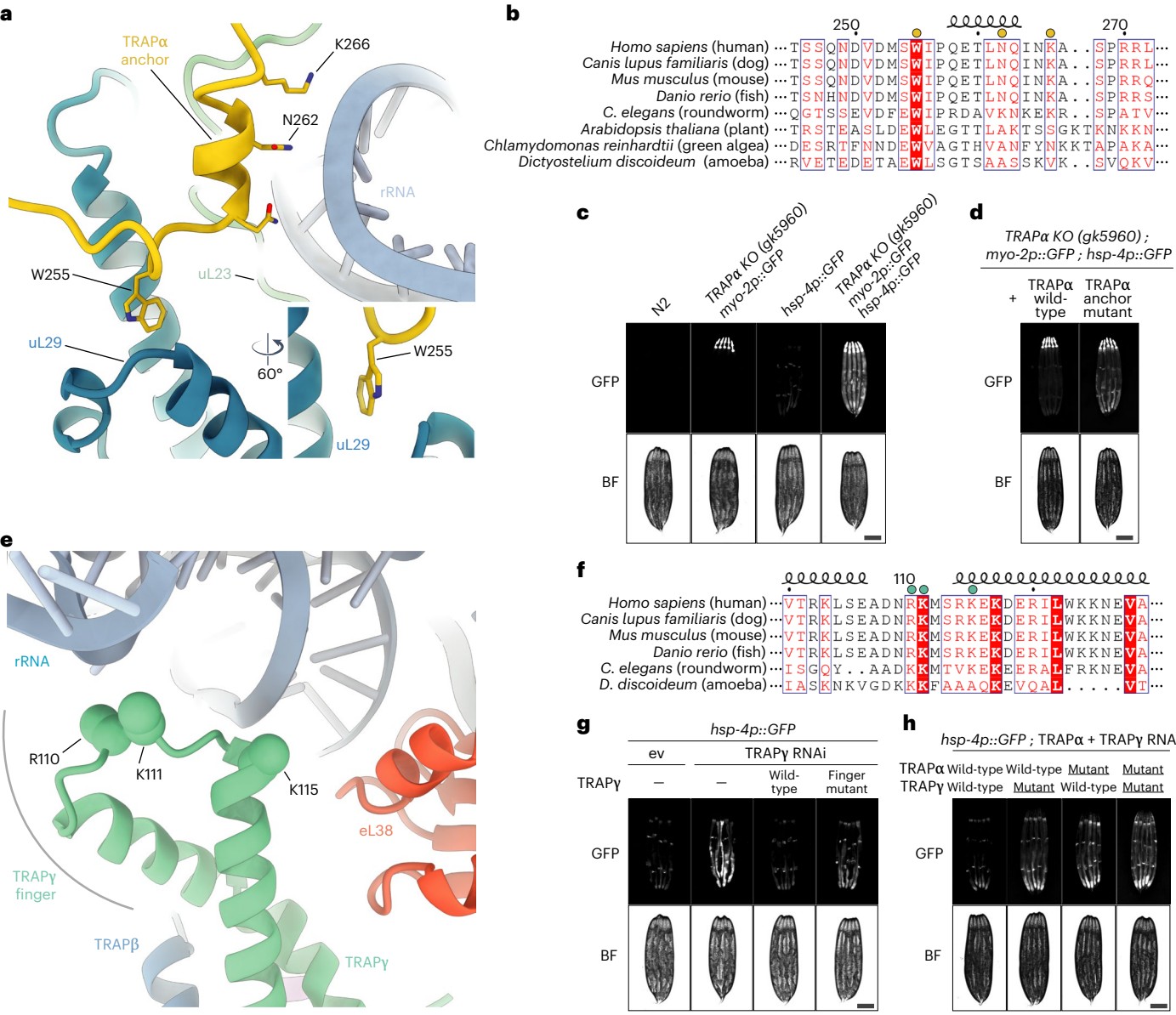

**Fig. 2 | Molecular interactions of TRAP with the translating ribosome.**
**a**, Atomic model of the TRAPα anchor. TRAPα (yellow) is shown in cartoon representation and the mutated residues that interact with ribosomal protein uL29 (blue) and ribosomal RNA (light blue) are indicated. **b**, Sequence alignment of TRAPα in eukaryotes. Mutated residues are indicated with a yellow dot above the sequence. **c**, Fluorescence microscope images of *C. elegans* TRAPα KO strain carrying a pharynx-specific myo-2p::GFP expression cassette in the TRAPα gene locus, and the ER stress reporter hsp-4p::GFP. Wild-type N2 strain served as negative control. Scale bar, 0.2 mm. **d**, Fluorescence microscope images of *C. elegans* TRAPα KO/hsp-4p::GFP strain as in **c** complemented either with TRAPα wild-type or anchor mutant. Scale bar, 0.2 mm. **e**, Atomic model of

the TRAPγ cytoplasmic domain. Positively charged residues of TRAPγ (green) that potentially interact with negatively charged ribosomal RNA (light blue) and were mutated are indicated. Ribosomal protein eL38 is shown in red. **f**, Sequence alignment of TRAPγ in eukaryotes. Mutated residues are indicated with a green dot above the sequence. **g**, Fluorescence microscope images of *C. elegans* expressing hsp-4p::GFP and carrying TRAPγ RNAi-resistant transgenes as indicated. Analysis was performed in the endogenous TRAPγ RNAi background on day 1 of adulthood. Scale bar, 0.2 mm. **h**, Similar analysis as in **g**, but in the endogenous TRAPα + γ RNAi background with strains expressing indicated RNAi-resistant TRAPα and TRAPγ transgenes. Scale bar, 0.2 mm. ev, empty vector RNAi control; BF, brightfield.

Considering that both the TRAPα anchor and the TRAPγ finger are important for TRAP interactions with the ribosome, we also investigated the levels of the ER stress reporter hsp-4p::GFP in *C. elegans* coexpressing the mutants of both proteins. As expected, ER stress was further enhanced in the double mutant worms, with the reporter hsp-4p::GFP now detected not only in the highly secretory intestinal cells but also in other tissues, such as muscle cells (Fig. 2h and Extended Data Fig. 9c). In addition to ER stress, TRAPα KO animals were notably smaller (Extended Data Fig. 9d), suggesting TRAP dysfunction causes a general growth defect in worms, consistent with

the critical role of TRAP in the biosynthesis and secretion of insulin-like growth factors in *C. elegans*[13,33]. Animals expressing the mutant TRAP subunits, particularly the TRAPα anchor mutant, also exhibited a substential growth defect, suggesting that the ribosome contact of the TRAP complex is essential for proper cell growth and animal development (Extended Data Fig. 9e).

These results suggest that the two ribosome contact points of TRAP have complementary roles. While the anchor provides affinity for TRAP, it is connected to the rest of TRAP via a long flexible linker and therefore does not restrain the position of TRAP relative to the

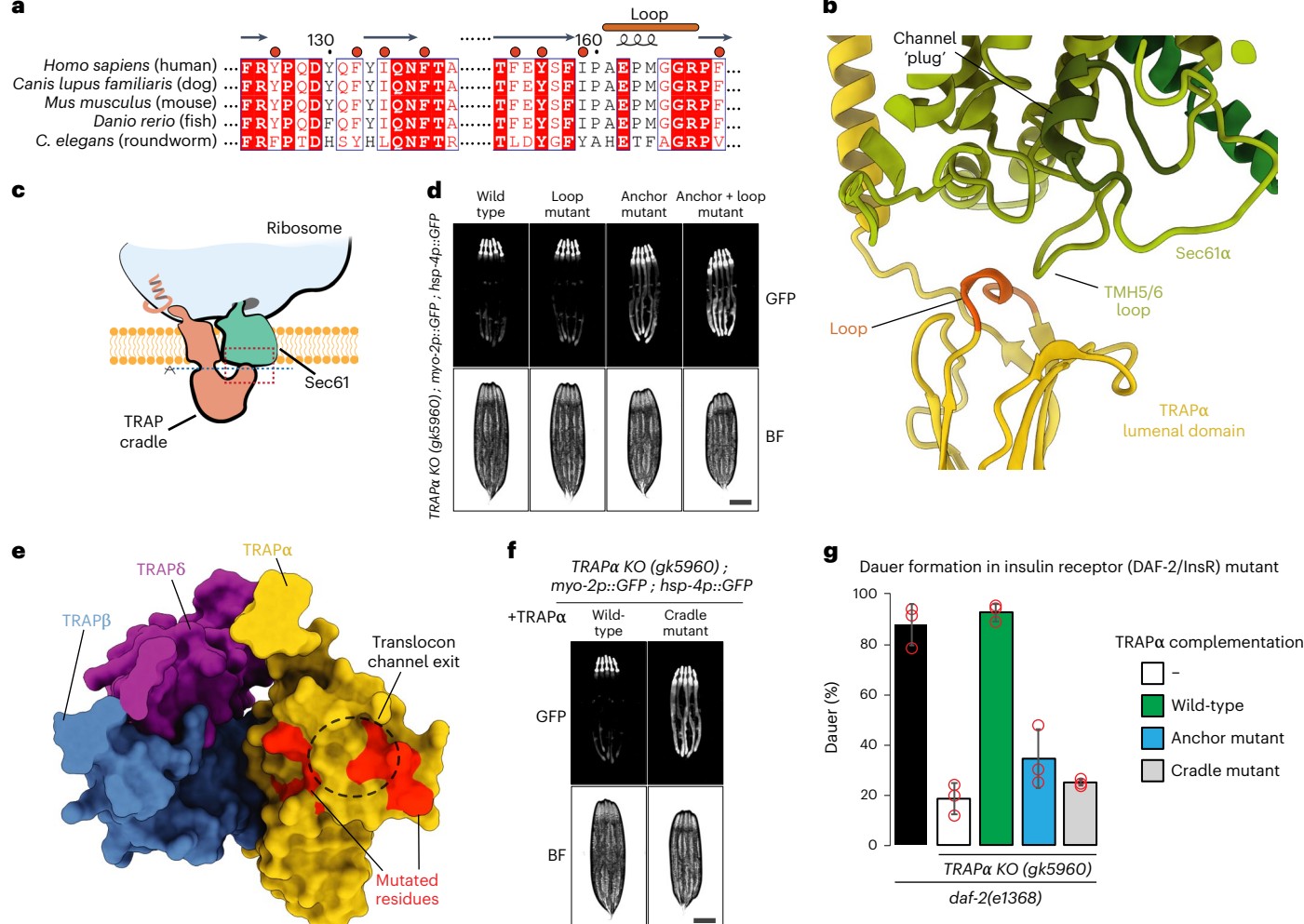

**Fig. 3 | Molecular interactions between the lumenal side of TRAP and the Sec translocon. a,** Sequence alignment of TRAPα in eukaryotes. Loop that potentially contacts the Sec translocon is marked. Hydrophobic residues located within the cradle that were mutated are indicated with a red dot above the sequence. **b,** Closeup of the contact region between TRAP and the Sec translocon. Region of the closeup is marked with a dashed red box on a schematic in **c.** Loop of TRAPα that potentially contacts Sec translocon is indicated and colored light brown. Translocon channel plug is labeled and shown in dark green. **c,** Schematic of the Sec translocon and TRAP complex bound to the ribosome. Dashed red box indicates the closeup region shown in **b.** Dashed blue line indicates the plane of view used in **e.** Ribosome, Sec translocon and TRAP cradle lumenal domain are labeled. **d,** Fluorescence microscope images of *C. elegans* TRAPα

KO worms expressing hsp-4p::GFP and indicated TRAPα variants. Analysis was performed on day 1 of adulthood. Scale bar, 0.2 mm. **e,** TRAP complex lumenal domain is shown as surface representation with proteins colored individually. Each TRAP protein that contributes to the lumenal domain is labeled. Sec translocon channel pore exit is indicated with a black dashed circle. Residues that were mutated are indicated in red. **f,** Fluorescence microscope images of *C. elegans* TRAPα KO worms expressing hsp-4p::GFP and complemented with either wild-type TRAPα or the cradle mutant. Scale bar, 0.2 mm. **g,** TRAPα KO worms expressing the indicated TRAPα variants and carrying the daf-2(e1368) mutation were grown at 24.5 °C for 2 days. Diagram shows percentage of worms in the dauer state. Data are presented as mean values ± s.d. *n* = 3 independent experiments. Red circles indicate individual datapoints.

ribosome or the translocon. In contrast, the cytosolic domain of TRAPγ contacts the ribosome via a finger-like loop flanked by α-helical elements and, therefore, together with the translocon contact of the lumenal domain of TRAPα (described below), helps in positioning the TRAP complex below the exit of the Sec61 channel.

## TRAPα lumenal domain interacts with translocated proteins

Our structural results also show that TRAPα contacts the lumenal loop of Sec61α next to the protein-translocating pore. Based on the structure presented here, a highly conserved loop between β sheets 5 and 6 of the lumenal domain of TRAPα (Fig. 3a) is in position to interact with the loop between TMHs 5 and 6 of the Sec61α subunit of the translocon (Fig. 3b,c). Mutation of the loop residues to polyserine (133HETFAGR/ SSSSSSS139, TRAPα loop mutant) in *C. elegans* markedly exacerbated the

ER stress phenotype and growth defect of the TRAPα anchor mutant (Fig. 3d and Extended Data Fig. 10). This suggests that the function of the TRAP complex relies on a dual binding mode of TRAPα interacting with both the translating ribosome in the cytosol and the translocating Sec61 pore in the ER lumen. Considering the positioning of TRAPα lumenal domain below the pore of the Sec translocon where it interacts with TRAPβ and TRAPδ to create a molecular cradle, it is reasonable to assume that it will interact with translocating nascent polypeptides. Analysis of the residues lining the surface of the cradle below the pore reveals the presence of numerous conserved aromatic and other hydrophobic residues in TRAPα, suggesting a possible role in interactions with the unfolded nascent polypeptides after passage through the translocon (Fig. 3a–c). The observed molecular arrangement and the surface features of the complex are reminiscent of the mode of interaction between the bacterial trigger factor chaperone,

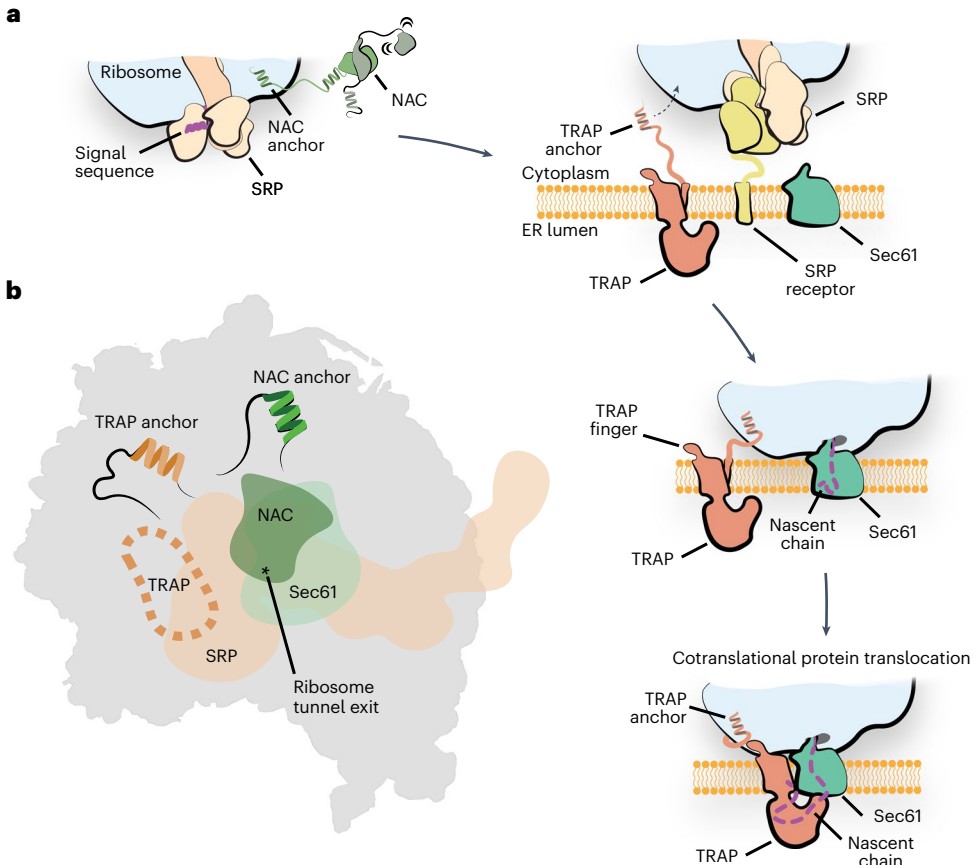

**Fig. 4 | Model of TRAP function in protein translocation across the ER membrane. a**, Schematic of the proposed function of the TRAP complex. The interplay of NAC and SRP initiates the targeting of the translating ribosome carrying an ER client to the ER membrane. Interaction with the SRP receptor at the ER membrane then initiates the transfer of the ribosome to Sec translocon. The tethered ribosome interacting TRAPα anchor could facilitate the transfer by attaching the ribosome near the Sec translocon. Once the handover has occurred, the TRAP-RNC-Sec61 complex is additionally stabilized by the TRAPγ

finger on the ribosome as well as the TRAPα lumenal loop interacting with the translocon. In this conformation, the TRAP complex positions a hydrophobic cradle-like lumenal domain directly at the exit of the Sec61 pore, which could act as a chaperone for the incoming nascent chain. **b**, Schematic depicting the binding sites of SRP, NAC and Sec translocon on the ribosome exit tunnel region. The binding site of the TRAP anchor and NAC anchor are indicated. Binding site of TRAP finger is indicated with a dashed line. The position of the ribosome polypeptide tunnel exit is indicated with an asterisk.

which binds to the translating ribosome to present a hydrophobic cradle to the nascent chains of cytosolic proteins[34,35].

To test the importance of the conserved hydrophobic amino acids on the inside of the TRAP cradle (Fig. 3a,e), we designed a *C. elegans* mutant in which these residues (F98-Y104-L106-F109-L126-Y128-Y131-V141) were mutated to threonines, a polar, uncharged amino acid (TRAPα cradle mutant). Worms expressing this mutant showed strongly increased expression levels of the ER stress reporter hsp-4p::GFP, especially in highly secretory intestinal cells (Fig. 3f and Extended Data Fig. 10a). In addition, these worms also showed a pronounced growth defect similar in severity to the TRAPα anchor mutant (Extended Data Fig. 10b). These results show that the cradle-shaped luminal domains of TRAP participate in the biogenesis of nascent chains as they are translocated into the ER lumen, possibly acting as a molecular chaperone. Similar domains exist in EMC, SPC22/23, OST and Hsp70 and were proposed to have a chaperone-like function based on their structural features[35–38].

Previous studies suggested that TRAPα is essential for insulin secretion in human cells and *C. elegans*[13,14]. To investigate whether the ribosome-binding anchor and putative substrate-interacting cradle of TRAPα are important for insulin secretion, we used a genetic model of *C. elegans* insulin secretion carrying a mutation in the insulin receptor DAF-2/InsR that results in enhanced dauer larvae formation due to an

insulin signaling defect[39,40]. *C. elegans* has an unusually complex insulin system and expresses 40 different insulin-like peptides, some of which enhance dauer arrest by antagonizing DAF-2/InsR signaling (for example, INS-1—the closest relative of human insulin)[41,42]. Consistent with a previous study, KO of TRAPα prevented dauer formation in DAF-2/InsR mutant animals, suggesting that secretion of insulin-like peptides that antagonize DAF-2/InsR depends on TRAP[13] (Fig. 3g). While expression of wild-type TRAPα fully restored dauer formation in TRAPα KO/DAF-2/InsR mutant animals, the TRAPα anchor and cradle mutant variants only showed minor activity, suggesting an insulin secretion defect in these animals (Fig. 3g). Defective secretion of other dauer-promoting factors unrelated to insulin could also contribute to the suppression of dauer formation in these mutants.

## Discussion

Based on the results presented here and previous insights into the structure and function of the TRAP complex[7,27], we propose a model for its participation in ER protein biogenesis (Fig. 4a). Initial interactions between ribosomes targeted to the ER and the TRAP complex occur via an evolutionarily conserved TRAPα anchor, which is flexibly tethered to the rest of TRAP. Considering that these contacts do not overlap with the position of SRP or NAC[21] (Fig. 4b and Extended Data Fig. 8) and that TRAP was observed to interact with nascent chains independent of the

Sec translocon[14], it is possible that TRAP anchor contacts contribute to targeting of ribosomes to the ER. Once the ribosome binds the translocon, TRAP fully engages the ribosome, stabilized by the additional electrostatic contacts with TRAPγ finger and interactions between the lumenal domain of TRAPα with the Sec61 translocon. In this conformation, a hydrophobic cradle formed of TRAPαβδ lumenal domains is positioned below the exit of the translocon pore for interactions with emerging nascent polypeptides. Our in vivo experiments in *C. elegans* show that the observed contacts and the hydrophobic character of the cradle, which possibly carry a chaperone-like functions, are critical for the biogenesis of secreted and membrane proteins in the ER.

Our results indicate that TRAPα fulfills two main functions of TRAP. It provides the anchor for the attachment to the ribosome as well as the hydrophobic domain that interacts with the Sec translocon and can aid in protein folding and biogenesis on the luminal side of the ER. It is therefore conceivable that the ancestral TRAP was a single domain protein that, over the course of evolution, acquired additional subunits, possibly in part through domain duplication as suggested by the similar domain folds of TRAPα, TRAPβ and TRAPδ (Fig. 1d,e), which helped stabilize the complex and optimize its positioning next to the exit of the translocon pore. This hypothesis is supported by the observation that a reduced functional TRAP complex, composed of only TRAPα and TRAPβ, exists in plants and algae[7]. This system would be able to fulfill all key roles of TRAP[43]; however, compared with the tetrameric mammalian TRAP, it would lack the additional stabilizing electrostatic interactions mediated by TRAPγ.

Considering its position relative to the RNC and the Sec translocon, it is likely that TRAP can act on nascent chains simultaneously with several other protein complexes known to participate in protein translocation, such as OST[24] or the factors involved in the formation of multipass membrane proteins, such as PAT complex or TMCO1 translocon[44,45]. However, it is also likely that TRAP may need to occasionally move in or out of its position next to the translocon to allow sequential access of a range of other nascent chain interacting factors in the ER, such as TRAM (translocating chain-associating membrane protein), signal peptidase complexes and EMC (endoplasmic reticulum membrane protein complex)[3,6]. Our results explain how TRAP would be able to accomplish this task through flexible anchor attachment. A similar mode of action has been observed recently for NAC, as it tethers SRP to control its access to the signal sequence-containing nascent chains[31].

The structural and in vivo approach used in this study allowed us to identify and dissect the importance of the TRAPγ key residues responsible for ribosome interactions. Previous cryo-electron tomography reconstructions observed these contacts; however, without Alphafold2 it was not possible to obtain a molecular model of the TRAP complex and experimentally validate these interactions[7]. Additionally, we discovered that the TRAPα C-terminal tail anchors the complex to the ribosome and its lumenal domain contacts the Sec translocon, and we showed that both of these interactions are important for TRAP function. The functionally important interaction between TRAPα anchor and the ribosome is not described in the two related manuscripts recently deposited with *BioRxiv*[46,47], in which the structure of the ternary complex of TRAP, Sec translocon and a ribosome was investigated. Finally, we showed that the hydrophobic residues located within the TRAP lumenal domain are critical for the biogenesis of nascent chains, suggesting a potential chaperone function. These results provide important insights into the molecular basis of TRAP participation in the biogenesis and translocation of proteins in the ER. Nevertheless, to fully understand its mechanism of action and substrate specificity in membrane protein biogenesis further experiments, including mutational studies with purified TRAP complex, will be critical.

## Online content

Any methods, additional references, Nature Portfolio reporting summaries, source data, extended data, supplementary information,

acknowledgements, peer review information; details of author contributions and competing interests; and statements of data and code availability are available at https://doi.org/10.1038/s41594-023-00990-0.

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

## Methods

### Preparation of mammalian RNC complex

The RNC was prepared and purified as previously described[21]. Briefly, a plasmid encoding 3×FLAG tag followed by an N-terminal fragment of the yeast dipeptidyl aminopeptidase B protein (24–90 amino acids)[48] was linearized using *Pst*I restriction enzyme and used for mRNA generation in in vitro transcription with the T7 RNA polymerase. The mRNA at a concentration of 214 ng $\mu l^{-1}$ was translated in the Flexi Rabbit Reticulocyte Lysate System (Promega) for 25 mins at 32 °C resulting in run-off RNC. The RNCs were purified using FLAG-tag affinity chromatography. Approximately 0.5 ml of ANTI-FLAG M2 Affinity Gel (Sigma-Aldrich) was washed with buffer A (50 mM HEPES-KOH pH 7.6, 100 mM KCl, 5 mM $MgCl_2$) and incubated with 4.7 ml of the translation reaction product for 2 h at 4 °C in chromatography column. The supernatant was then removed by gravity flow and the gel was washed with 10 ml of buffer B (50 mM HEPES-KOH pH 7.6, 500 mM KCl, 5 mM $MgCl_2$) and 10 ml of buffer A (50 mM HEPES-KOH pH 7.6, 100 mM KCl, 5 mM $MgCl_2$). The RNCs were eluted in three fractions (1 ml each) with buffer A containing 0.1 mg $ml^{-1}$ 3×FLAG peptide. All fractions were pooled and the RNCs were pelleted by ultracentrifugation in a TLA55 rotor (Beckman Coulter) at 153,587*g* at 4 °C for 2 h. Finally, the RNC pellet was resuspended in buffer C (50 mM HEPES-KOH pH 7.6, 100 mM KOAc, 5 mM Mg(OAc)₂) to a final concentration of 360 nM. The sample was flash-frozen in liquid nitrogen and stored at –80 °C.

### Cryo-EM sample preparation

The mammalian RNCs were incubated with canine SRP (tRNA Probes LLC) at a final concentration of 100 nM RNC and 140 nM SRP in a buffer R1 (50 mM HEPES-KOH pH 7.4, 5 mM $MgCl_2$, 150 mM KOAc, 300 mM sucrose, 5 mM guanosine-5′-[(β,γ)-imido]triphosphate (GNP) for 10 min at 30 °C. Concurrently, 20 µl of EKRM (tRNA Probes LLC) were incubated for 5 min on ice in buffer R2 (50 mM HEPES-KOH pH 7.4, 5 mM $MgCl_2$, 150 mM KOAc, 440 mM sucrose, 5 mM GNP, 1 mM RNaseOUT). Afterwards, both reactions were mixed together and incubated for 10 min at 30 °C. The EKRM were then solubilized with 2% digitonin for 15 min at 4 °C and the reaction was then centrifuged for 10 min at 12,000*g* and 4 °C. The supernatant was carefully layered onto 100 µl of buffer C (50 mM HEPES-KOH pH 7.4, 5 mM $MgCl_2$, 150 mM KOAc, 0.025% GDN, 1.4 M sucrose) and the sucrose cushion was ultracentrifuged for 1.5 h at 278,088*g* and 4 °C in TLA-100 rotor (Beckman Coulter). The pellet was resuspended in 100 µl of buffer F (50 mM HEPES-KOH pH 7.4, 5 mM $MgCl_2$, 150 mM KOAc, 0.025% GDN, glycodiosgenin) and spun down again for 10 min at 12,000 rcf and 4 °C. The final sample in the supernatant was transferred into a fresh tube and use immediately for preparation of cryo-EM grids.

### Cryo-EM grid preparation and data collection

Quantifoil R2/2 holey carbon grids were washed with ethyl acetate, coated with an extra layer of carbon and glow discharged with 15 mA for 15 s using the Pelco EasyGlow system. Each grid was mounted onto the ThermoFisher Vitrobot IV and 3.5 µl of sample was incubated on the grid for 60 s at 4 °C and 100% humidity before being blotted and plunged into liquid ethane/propane mix cooled to liquid nitrogen temperature. Several grids with different blotting times were prepared from the sample at approximate 400 nM, 200 nM and 100 nM ribosome concentration (based on absorbance at 260 nm wavelength).

Data were collected on Titan Krios electron microscope operated at 300 kV and equipped with the Gatan K3 direct electron detector and Gatan Imaging Filter with an energy filter slot width of 20 eV. Automated data acquisition in counting mode was performed using EPU software. Data were collected at a nominal magnification of ×81,000 and a defocus range of –1.2 to –3 µm, with a pixel size of 1.06 Å per pixel. Micrographs were recorded as movie stacks with an electron dose of ~60 e/Å².

### Cryo-EM data processing

A total of four datasets were collected from two different cryo-EM grid preparations. Two data collections, dataset 1.1 and dataset 1.2, containing 9,373 and 25,548 movies, respectively, were collected from the same grid preparation. These two datasets were then merged into dataset 1 and imported into RELION v.3.1[49] with separate optics groups. Dataset 2 was composed of dataset 2.1 (6,628 movies) and dataset 2.2 (23,235 movies) prepared from the second grid preparation. Both datasets were motion corrected with MotionCorr2[50] and the micrographs contrast transfer function (CTF) was estimated using CTFFind4[51]. Particles were picked in RELION using 80S ribosome as a reference. Extracted particles (binned, at a pixel size of 6.784 Å) were subjected to two-dimensional (2D) image classification. Particles from 2D class averages that depict well-resolved ribosomes were picked from both datasets, merged together and subjected to another round of 2D classification. Classified particles were selected into one of three groups according to their class averages: ribosome particles, ribosome-like particles and protein-like particles. Each group was subjected to three rounds of 3D classification with an 80S ribosome as a reference lowpass filtered to 60 Å resolution. Selected 610,191 ribosomal particles were re-extracted with a 1.428 Å per pixel and a box of 320 × 320 pixels and were 3D autorefined using lowpass filtered 80S ribosome as a reference, with per-particle CTF and aberration correction. Focused 3D classification was then performed with a mask surrounding the exit tunnel of the ribosome and the micelle region. This was used to improve the density for the TRAP and Sec translocon complexes. Classification was performed without particle alignments as described in RELION[49], with regularization parameter $T = 3$ and limiting the resolution in the E-step to 5 Å. The class showing a strong density for the detergent micelle and an extra lumenal domain of TRAP and Sec translocon was selected. A second round of focused 3D classification to further improve the occupancy of the TRAP complex. A class composed of 114,154 particles with a strong density of the Sec translocon was selected and 3D autorefined. A final round of focused 3D classification was performed using a mask surrounding the TRAP complex only, with a regularization parameter $T = 10$, E-step limit resolution of 6 Å. The class depicting the best-resolved density for the TRAP complex was selected containing 22,643 particles, re-extracted without rescaling (pixel size of 1.06 Å) and was then 3D refined to a global resolution of 3.5 Å.

### Model building

A recently published high-resolution model of the rabbit 80S ribosome (PDB 7O7Y)[52] was docked into the cryo-EM map using UCSF ChimeraX[53] and readjusted manually in COOT[54]. This included repositioning of the large subunit stalks as well as residues lining the nascent chain tunnel and the tunnel exit, where the ribosome contacts the Sec61 translocon. Coordinates for the Sec61 translocon were used based on the PDB 6W6L[45] and rebuilt in the areas where the map reached near-atomic resolution (Extended Data Fig. 4). For building a P-site acyl-tRNA (His) template together the attached nascent chain, a recently published high-resolution cryo-EM map of the same ribosome nascent chain complex (EMD-12801) served as a guide, and the resulting model was transplanted into the current map. The sequence for the tRNA (tRNA-His-GTG-1-1) was obtained from the GtRNAdb database[55]. An initial molecular model of the tetrameric TRAP complex was predicted with AlphaFold2 in multimer mode[29,30]. The model was docked into the cryo-EM map using UCSF ChimeraX, and discrete parts (the lumenal domains of TRAPαβδ, the transmembrane domain bundle comprising TRAP and the transmembrane helix of TRAPα) were readjusted by rigid body fitting in COOT. The high quality of the EM map corresponding to the C-terminal TRAPα anchor allowed unambiguous sequence assignment and de novo building (Fig. 1c).

The assembled model was subjected to five cycles of real space refinement using PHENIX v.1.20.1[56] including side chain rotamer and Ramachandran restraints (Table 1). The model geometry was

**Table 1 | Cryo-EM data collection, refinement and validation statistics**

| | 80S nascent chain complex with TRAP and Sec61 (EMD-16232, PDB 8BTK) |
|---|---|
| **Data collection and processing** | |
| Magnification | ×81,000 (nominal) |
| Voltage (kV) | 300 |
| Electron exposure (e⁻/Å²) | 60 |
| Defocus range (µm) | 0.6–3.0 |
| Pixel size (Å) | 1.06 (super-resolution pixel at 0.53 Å per pixel) |
| Symmetry imposed | C1 |
| Initial particle images (no.) | 2,621,009 |
| Final particle images (no.) | 22,643 |
| Map resolution (Å) | 3.5 |
| FSC threshold | 0.143 |
| Map resolution range (Å) | 2.5–15 |
| **Refinement** | |
| Initial model used (PDB code) | 7O7Y, 6W6L |
| Model resolution (Å) | 3.6 |
| FSC threshold | 0.5 |
| Map sharpening B factor (Å²) | 20 |
| Model composition | |
| Nonhydrogen atoms | 233,722 |
| Protein residues | 13,271 |
| Nucleotides | 5888 |
| B factors (Å²) | |
| Protein | 21.8/457.9/168.1 |
| Nucleotides | 21.5/947.0/193.3 |
| Ligand | 1.5/482.1/105.0 |
| Root mean squared deviations | |
| Bond lengths (Å) | 0.001 |
| Bond angles (°) | 0.341 |
| Validation | |
| MolProbity score | 1.33 |
| Clashscore | 5.94 |
| Poor rotamers (%) | 0.66 |
| Ramachandran plot | |
| Favored (%) | 98.9 |
| Allowed (%) | 1.09 |
| Disallowed (%) | 0.01 |

validated using MolProbity[57]. The refined model shows an excellent geometry and map correlation, and the resolution of the model versus map Fourier shell correlation (FSC) at a value of 0.5 coincides well with that determined between the map half-sets at a FSC = 0.143 criterion (Table 1).

## In vivo experiments

***C. elegans* strains and transformation.** *C. elegans* worms were cultured according to standard techniques with *Escherichia coli* OP50 as food source[58]. ER stress reporter strain SJ4005 (zcIs4[hsp-4p::GFP])[32], TRAPα KO strain VC4892, in which the TRAPα gene is replaced by a selection cassette (gk5960[loxP + myo-2p::GFP::unc-54 3′ UTR + rps-27p::neoR::unc-54 3′ UTR + loxP])[59], and insulin receptor mutant strain DR1572 (daf-2(e1368) III.) were obtained from the Caenorhabditis Genetics Center (CGC, University of Minnesota, USA). Wild-type Bristol N2 strain was used for all transformations. Transgenic strains were generated using standard microinjection protocols[60]. Transgene integration was performed using the miniMos transposon method[61]. Strains carrying RNAi-resistant genes of TRAPα and TRAPγ were constructed as previously described[62]. In brief, RNAi-resistant coding sequences of *C. elegans* TRAPα (*trap-1*) and TRAPγ (*trap-3*) were designed using a codon adaptation tool[63] and synthesized by Integrated DNA Technologies (IDT). The coding sequences including three synthetic introns and a C-terminal FLAG tag are listed in Supplementary Tables 1 and 2. The genes were subcloned into miniMos pCFJ910 vector (Addgene plasmid catalog no. 44481)[61] under the control of the endogenous *trap-1* and *trap-3* promoter regions and 3′ UTRs. A separate fluorescent marker gene (mCherry) was added to the constructs to identify knock-in animals. Detailed strain information is available in Supplementary Table 3.

**ER stress reporter analysis.** Worm strains carrying RNAi-resistant TRAP genes were mated to the ER stress reporter strain SJ4005 expressing GFP under control of the ER stress-inducible *hsp-4* promoter (hsp-4p::GFP)[32]. Single and double RNAi constructs targeting endogenous *trap-1* and *trap-3* were cloned by inserting the spliced coding sequences of *trap-1* and *trap-3* into vector L4440 (Addgene plasmid catalog no. 1654). The constructs were then transformed into the RNAi feeding *E. coli* strain HT115[64]. Endogenous TRAP genes were silenced in worms from hatch on plates containing the respective HT115 RNAi bacteria. Worms were grown on RNAi plates at 20 °C until adulthood. Adult animals were then immobilized with 1% sodium azide and GFP fluorescence was assessed using a DM6000B-Cs microscope (Leica) equipped with a DFC 365FX camera (Leica) and a ×5 objective. Strains carrying TRAPα genes and hsp-4p::GFP were additionally analyzed in the TRAPα KO background without performing RNAi by crossing with the TRAPα KO strain VC4892, in which GFP is expressed constitutively in the pharynx (myo-2p::GFP). ER stress was assessed similarly in adult worms grown at 20 °C but with *E. coli* OP50 as food source. Each experiment was repeated independently three times with similar results, and representative images are shown.

**Worm growth analysis.** Strains carrying RNAi-resistant TRAPα and TRAPγ genes were grown on RNAi plates from hatch until adulthood at 20 °C to silence expression of the endogenous TRAP genes. Twenty gravid adult worms of each strain were then allowed to lay eggs on a fresh RNAi plate for 4 h. Adult worms were removed, and plates kept at 25 °C. The development of the worms was analyzed after 24, 48 and 72 h by determining the axial length of the worms (time-of-flight, TOF) by worm flow cytometry using a COPAS FlowPilot (Union Biometrica).

**Dauer arrest assay.** Dauer arrest assays were performed with *C. elegans* strain DR1572 (daf-2(e1368) III) as previously described[13]. In brief, gravid hermaphrodites were transferred to a fresh assay plate for synchronized egg lay at 20 °C. Adult animals were removed, and plates were transferred to 24.5 °C and scored for dauers after 2 days. The experiment was performed three times.

**Immunoblotting and antibodies.** Expression levels of the TRAP knock-in genes were analyzed by detection of the C-terminal FLAG tag using standard immunoblotting techniques as described previously[65]. Following commercial antibodies were used (Supplementary Information Table 4): FLAG (Sigma, catalog no. F7425), GAPDH (Proteintech, catalog no. 60004-1-lg). Tubulin antibodies were a kind gift from T. Mayer, University of Konstanz.

**Reporting summary**

Further information on research design is available in the Nature Portfolio Reporting Summary linked to this article.

## Data availability

The data supporting the findings of this study are available in the Electron Microscopy Bank and Protein Data Bank under accession codes EMD-16232 and PDB ID 8BTK. The structures of rabbit 80S ribosome (PDB 7O7Y) and Sec61 translocon (PDB 6W6L) were used for comparisons and as an initial model. Source data are provided with this paper.

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

## Acknowledgements

We thank members of the Ban and Deuerling laboratories for discussions. We thank A. Scaiola for help with EM data collection. We thank R. Schloemer and G. Hunaeus for technical assistance. Cryo-EM was collected at ScopeM at the ETH Zurich. This work was supported by the Swiss National Science Foundation (grant no. 310030_212308), National Center of Excellence in Research RNA and Disease Program of the SNSF (grant number 51NF40_205601) and, in part, by the Roessler Prize, Ernst Jung Prize, and Otto Naegeli Prize for Medical Research to N.B., by research grants from the German Science Foundation (SFB969/A01, A07, and C10) to E.D. and M.G., and by the School of Medicine at the University of Virginia and the Department of Molecular Physiology and Biological Physics startup funds to A.J.

## Author contributions

A.J., M.J. and N.B. conceived the study. M.J., A.J., M.G., E.D. and N.B. designed the experiments and analyzed the results. A.J. and M.J. prepared RNCs, assembled the ER-bound complex and prepared samples for cryo-EM experiments. M.J. and A.J. collected and processed cryo-EM data. M.J., A.J. and M.L. built and refined the atomic models. M.G. and S.S. made the *C. elegans* strains and did the in vivo experiments. M.J., A.J., M.G., E.D. and N.B. wrote the manuscript. All authors contributed to the final version of this manuscript.

## Funding

## Competing interests

The authors declare no competing interests.

## Additional information

**Extended data** is available for this paper at https://doi.org/10.1038/s41594-023-00990-0.

**Correspondence and requests for materials** should be addressed to Ahmad Jomaa, Elke Deuerling or Nenad Ban.

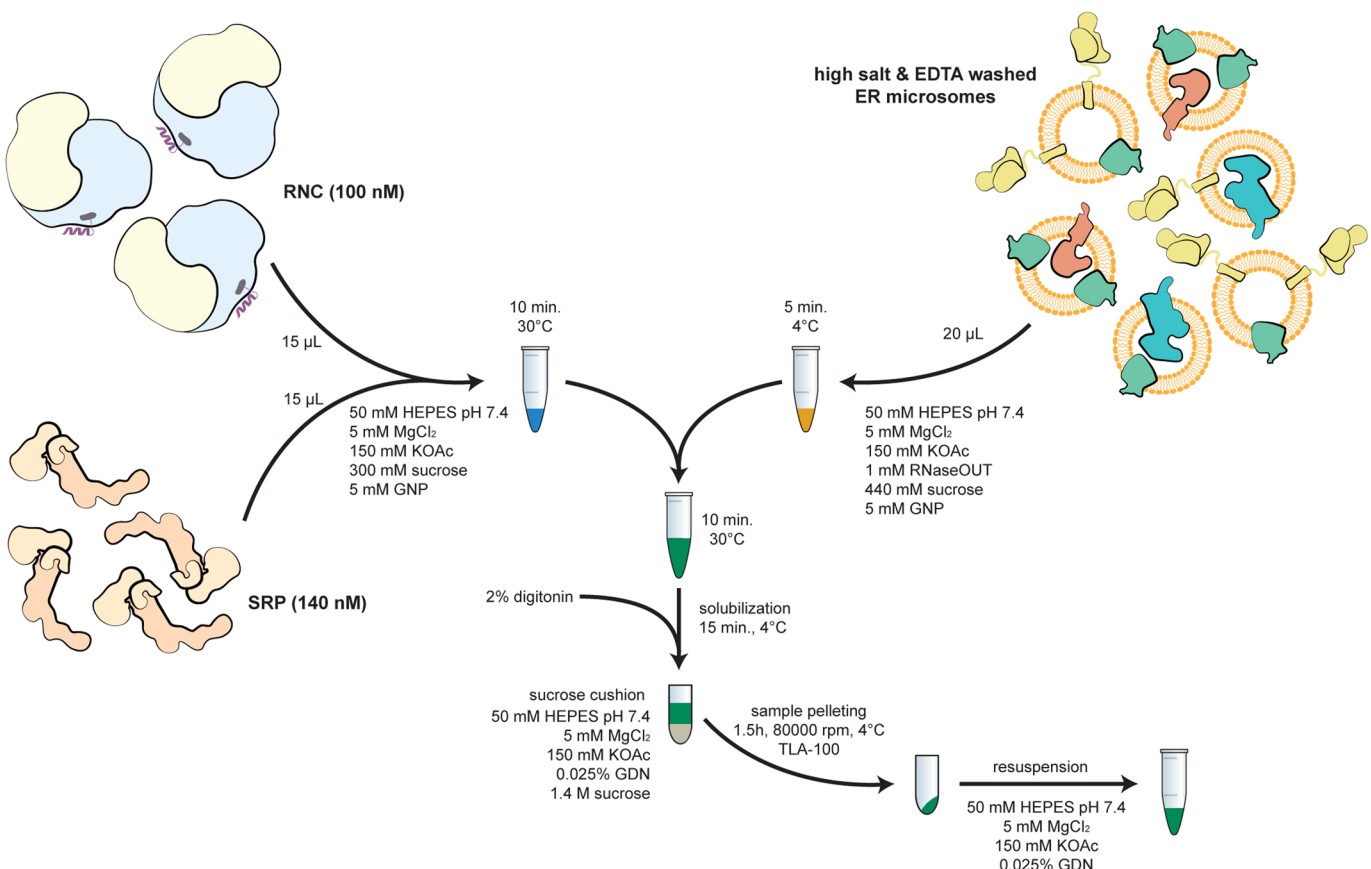

**Extended Data Fig. 1 | RNC-Sec61-TRAP complex assembly.** The complex was assembled using stalled RNCs carrying an ER client mixed with canine SRP and salt- and EDTA-treated canine ER microsomes (EKRM). Microsomes were first pelleted then solubilized with 2% digitonin. Ribosome-bound complexes were then pelleted over a sucrose cushion and applied directly to cryo-EM grids.

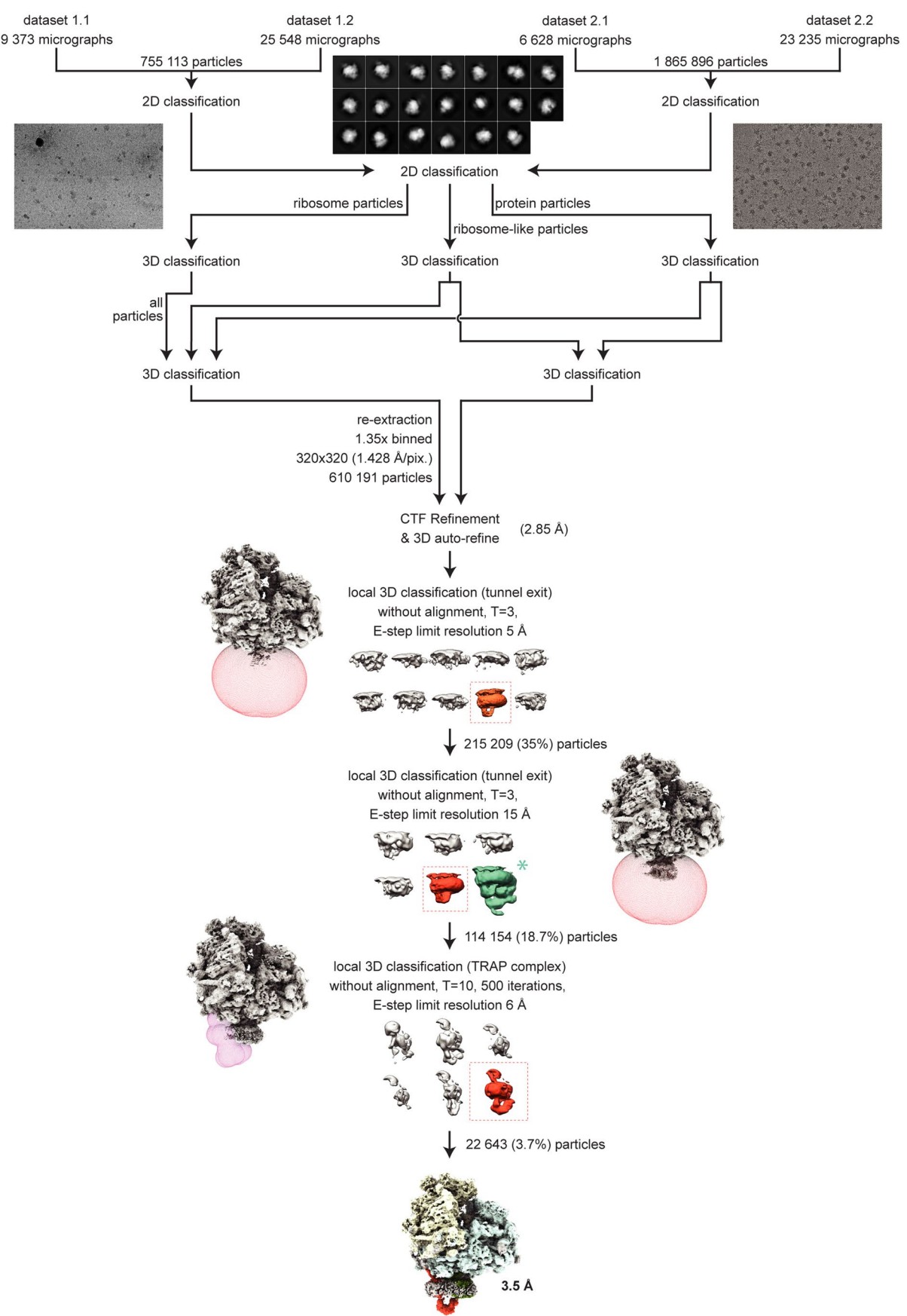

**Extended Data Fig. 2 | See next page for caption.**

**Extended Data Fig. 2 | Cryo-EM processing scheme.** Particles were picked from motion-corrected and dose-weighted micrographs in RELION 3.1 using lowpass filtered 80 S ribosome as a reference. Particles were then exposed to two rounds of 2D classification and three rounds of 3D classification to obtain ribosome particles depicting density for the stalled P-site tRNA and the Sec61 translocon. To improve first the density of Sec61 and then TRAP complex, particles were subjected to two round of focused 3D classifications with masks applied around Sec61 and the TRAP complex. Particles (22,643) from a 3D class that depicted the strongest density for TRAP and Sec61 were refined to 3.5 Å resolution. Green asterisk indicates the complex containing OST-Sec61-TRAP complex that co-purified in the sample and was not further discussed.

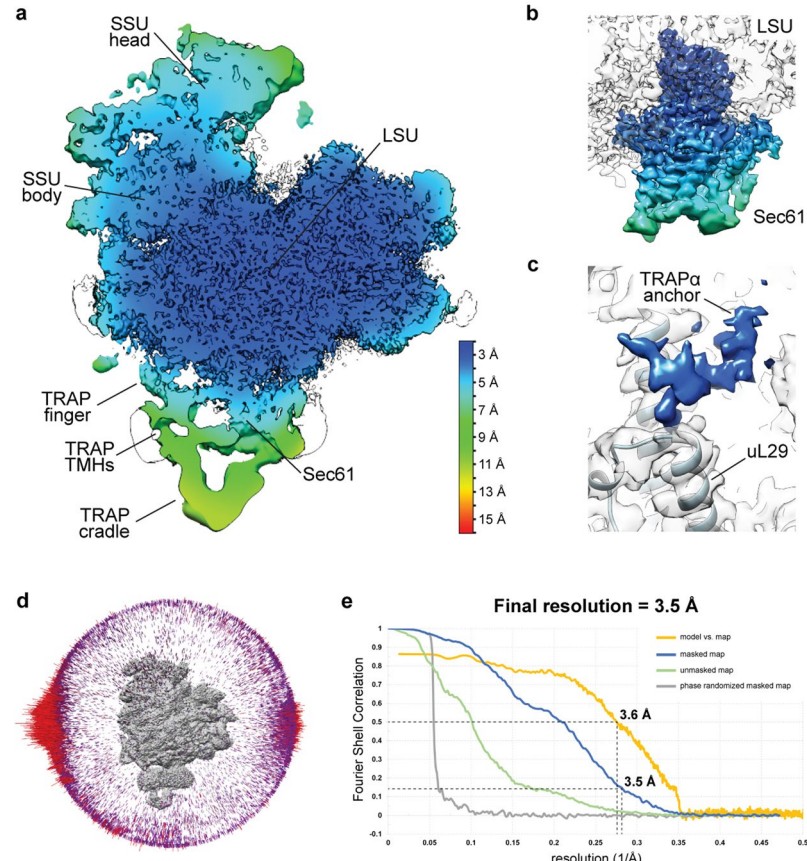

**Extended Data Fig. 3 | Local Resolution and Validation Plots. a,** Local resolution of the RNC-Sec61-TRAP complex calculated in RELION 3.1. Locally filtered map is showed (at 2.65σ), colored according to the resolution value. The scale bar is shown in the bottom right. **b,** Local resolution of the Sec61 translocon. Sec translocon map is shown at 2.4σ. Colors correspond to the same resolution as in panel (a). **c,** Local resolution of the TRAPα anchor. The map is shown at 3.6σ. For clarity, ribosome density was not colored according to the local resolution. Ribosomal protein uL29 that interacts with TRAPα anchor is indicated. **d,** Angular distribution of the particles in the two cryo-EM maps after final 3D refinements. **e,** Fourier Shell Correlation (FSC) plots and the model versus map plot, calculated using the gold standard FSC criteria cutoff (FSC = 0.143) using independent two half maps as implemented in RELION 3.1, and the cutoff for the resolution of the model is determined based on the FSC cutoff (FSC = 0.5).

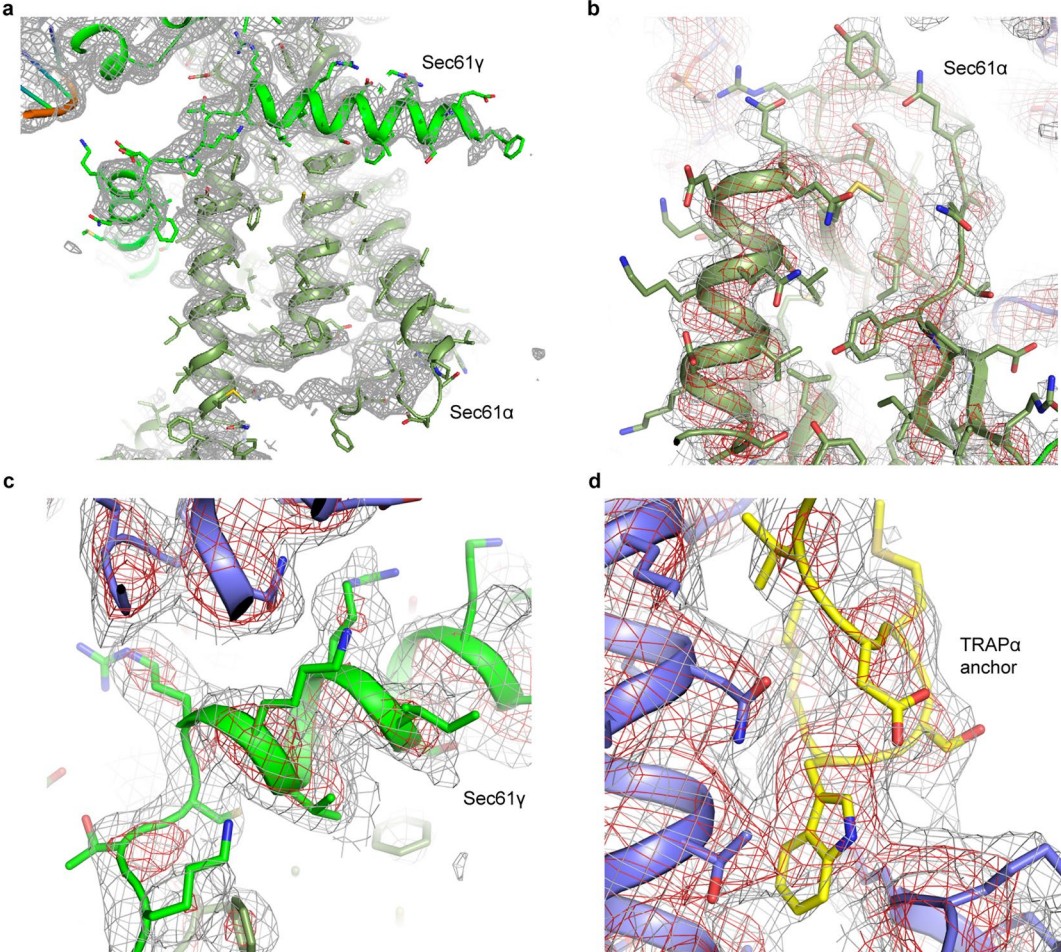

**Extended Data Fig. 4 | Cryo-EM density for the high-resolution areas of Sec61 and TRAP complexes. a–c**, Cryo-EM densities corresponding to the transmembrane α-helices of the Sec61α and Sec61α subunits. Cryo-EM density is shown as mesh and shown at two different contour levels and colored gray and red. Atomic coordinates are show as cartoon and sticks. Maps shown as gray mesh at 3.7σ (**a**), 4σ (**b**) and 3.3σ (**c**). **d**, Cryo-EM density corresponding to the TRAP anchor. Gray mesh shown at 3.5σ.

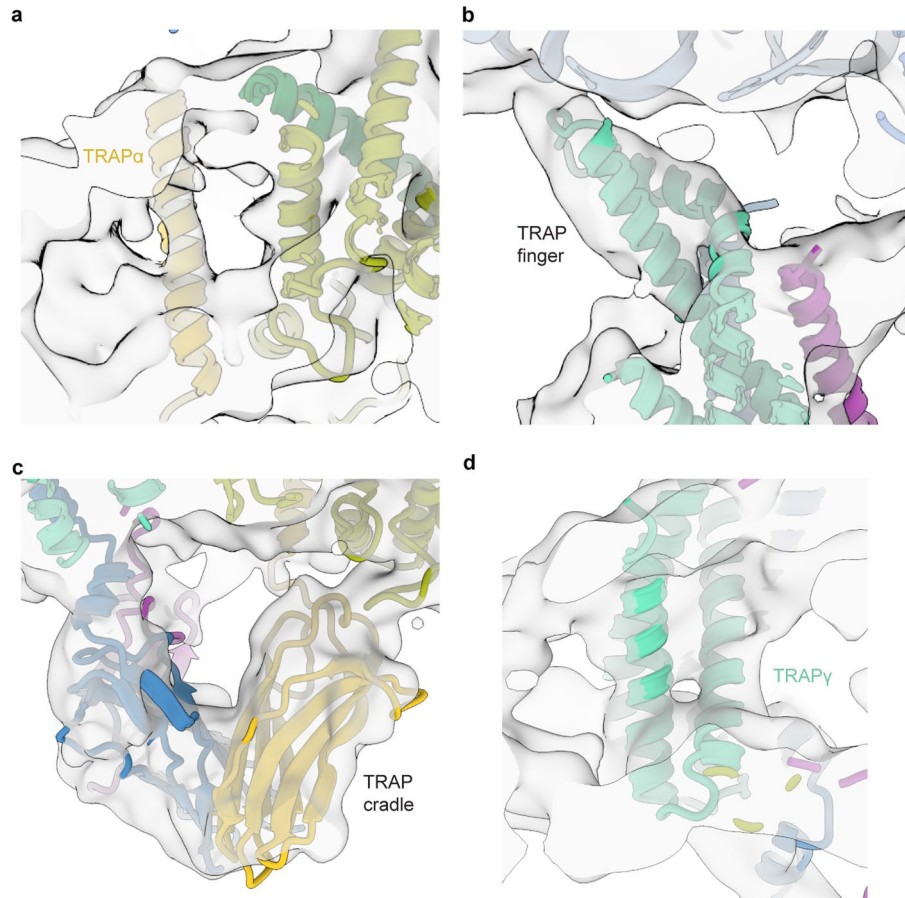

**Extended Data Fig. 5 | Closeups of the TRAP complex. a–d**, Closeup cryo-EM densities of the TRAPα transmembrane helix (**a**), TRAP finger (**b**), lumenal cradle domain (**c**) and TRAPγ transmembrane helices (**d**) overlaid with atomic coordinates. The map was filtered to 8 Å and shown at 3.1σ.

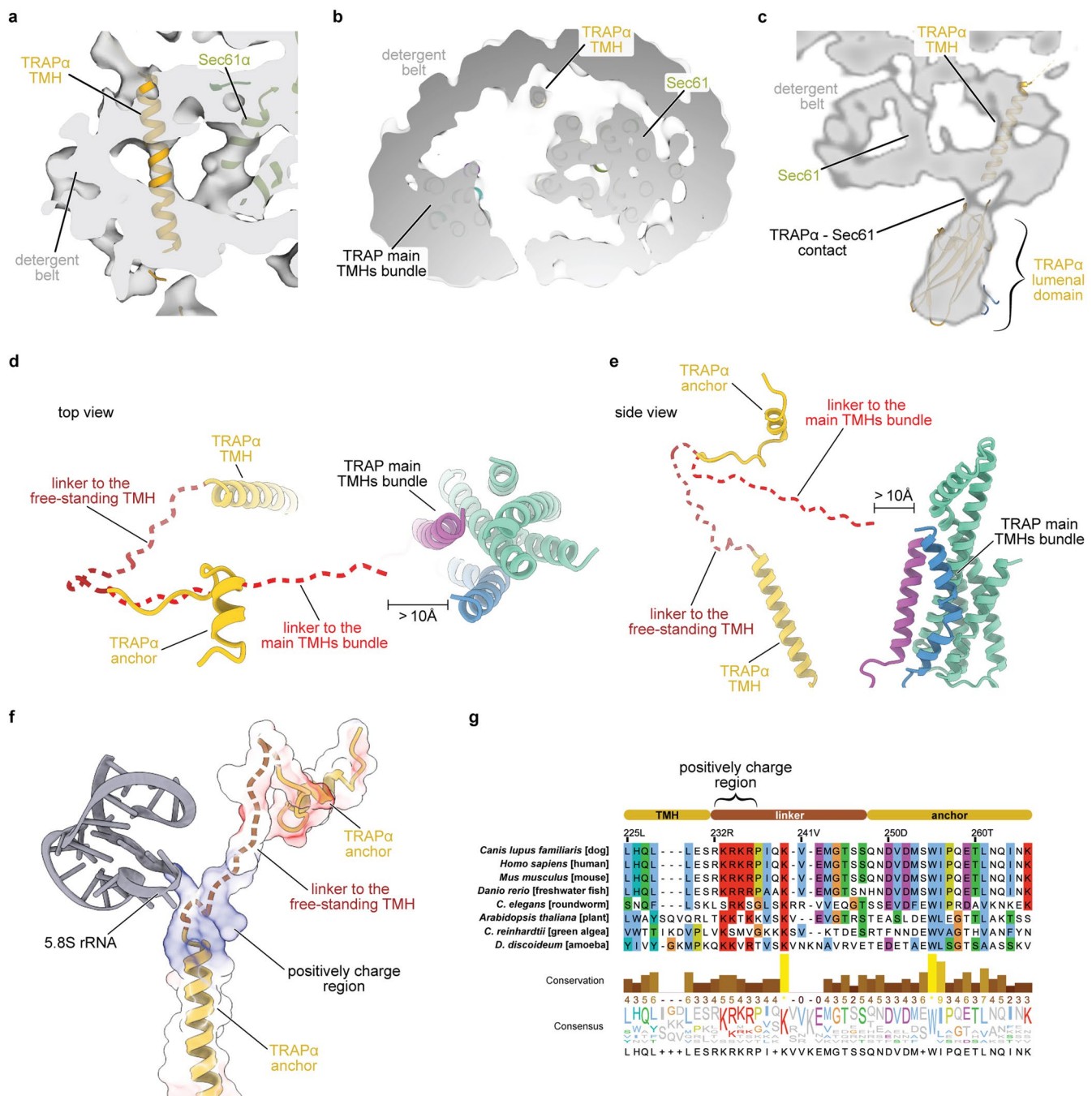

**Extended Data Fig. 6 | Placement of the free-standing TRAPα transmembrane helix. a, b,** Cross section of the detergent micelle in a side (**a**) and top (**b**) view. Sec61 translocon, TRAP complex main TMHs and the free-standing TRAPα TMH are indicated. The map was filtered to 7 Å and shown at 3.1σ. **c,** A solid plane of the cryo-EM map with underlaid TRAPα atomic model shown in yellow. **d, e,** Atomic model of the TRAP complex presented in this study with a potential path of the linker modelled. Brown dashed line represents a potential path of the linker to the free-standing TMH, whereas the linker would not be able to span the distance

to the TRAP complex main TMHs bundle. In the top view, TRAPγ residues 88–128 (cytoplasmic domain) are not shown for image clarity. TRAP complex proteins are colored as in Fig. 1. **f,** Part of the 5.8 S rRNA (gray) and TRAPα (yellow) shown in cartoon representation. A potential path of the linker is modelled and shown as brown dashed line. Electrostatic potential of TRAPα was calculated in ChimeraX and is shown as a transparent surface representation. **g,** Sequence alignment of TRAPα in eukaryotes with key elements of the TRAPα marked. Highly conserved positively charge region is indicated.

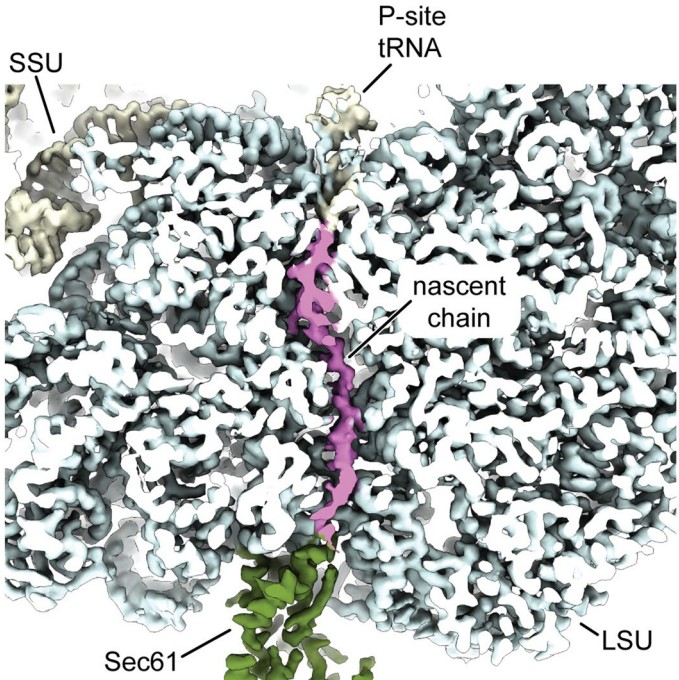

**Extended Data Fig. 7 | Closeup view of the ribosome tunnel with nascent chain.** Cross section of the ribosome showing the polypeptide exit tunnel. Cryo-EM density corresponding to the nascent chain is colored in magenta. Cryo-EM density corresponding to the Sec61 complex is colored green. Map shown at 3σ.

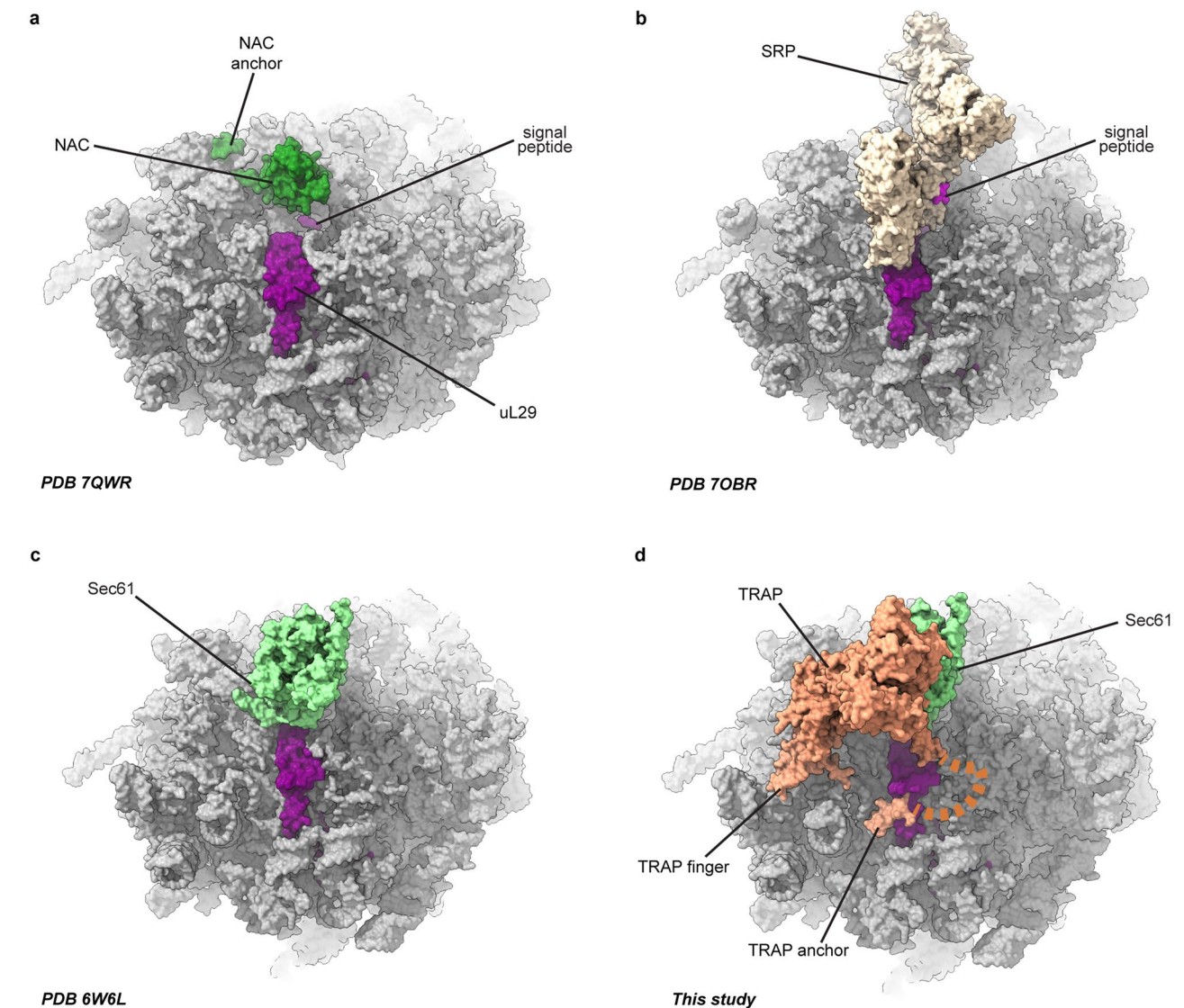

**Extended Data Fig. 8 | Comparison of factors positioning at the ribosome tunnel exit.** Closeup of the ribosome tunnel exit region of the RNC-NAC (PDB 7QWR), RNC-SRP (7OBR), RNC-Sec61(6W6L) and RNC-Sec61-TRAP (this study). Atomic coordinates are shown as surface representation.

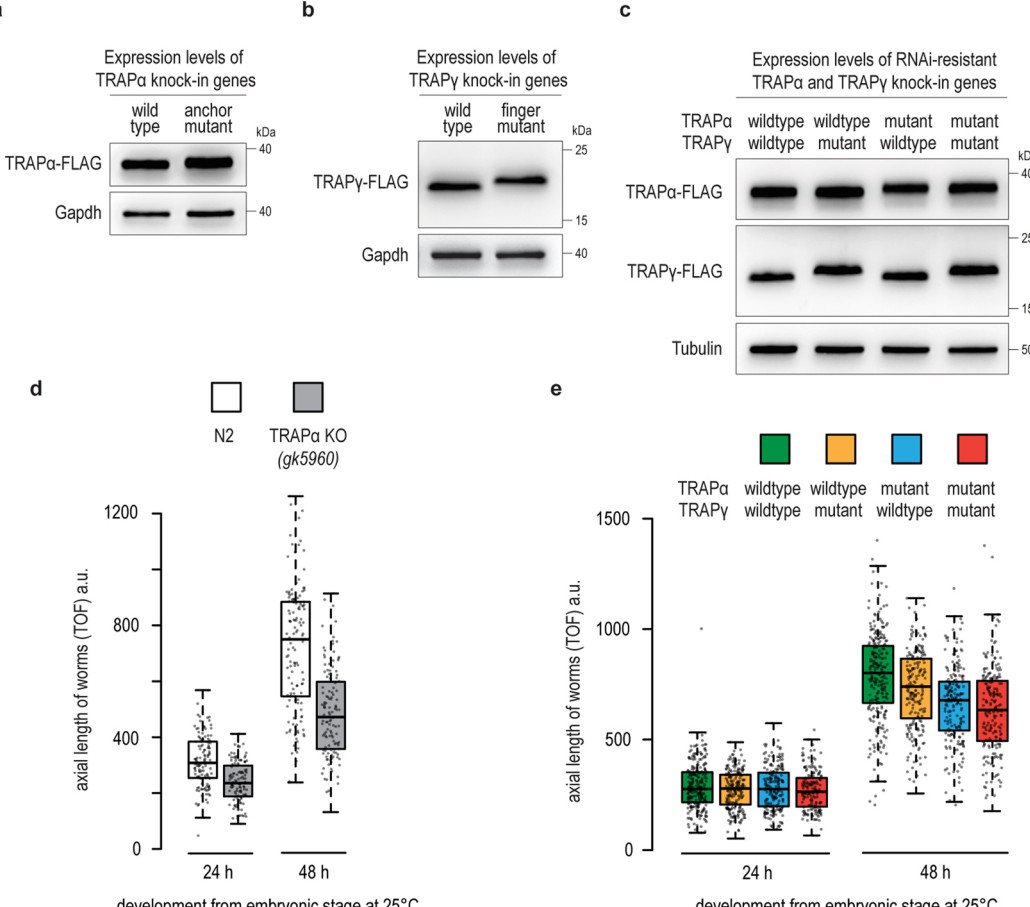

**Extended Data Fig. 9 | Characterization of TRAPα and TRAPγ transgenic *C. elegans* strains. a–c**, FLAG immunoblot analysis showing the TRAPα and TRAPγ expression levels in worms analyzed in main Fig. 2d,g, and h, respectively. All TRAP variants were tagged with a C-terminal mono-FLAG tag. Gapdh or Tubulin served as loading control. **d**, Growth analysis of wildtype N2 and TRAPα knockout worms. Worms were synchronized via a timed egg-lay and plates incubated for the indicated time at 25 °C. The axial length of worms was determined by worm

flow cytometry using a COPAS Biosorter. Box plot center line indicates the median, box length the upper and lower quartile, and whiskers the minimum/ maximum quartile (n = 125). Dots represent individual data points. TOF, time-of-flight. a.u., arbitrary units. **e**, Similar analysis as in panel d with worms expressing indicated RNAi-resistant TRAPα and TRAPγ transgenes (n = 175). Analysis was performed in the endogenous TRAPα and TRAPγ RNAi background (F2 RNAi generation).

**a**

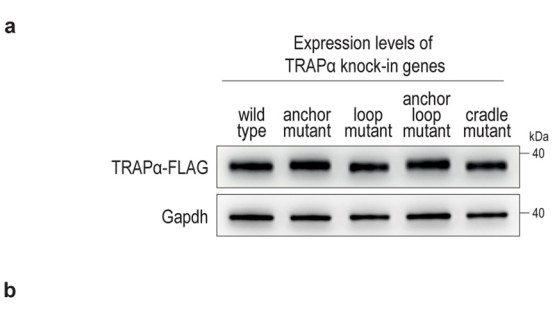

**b**

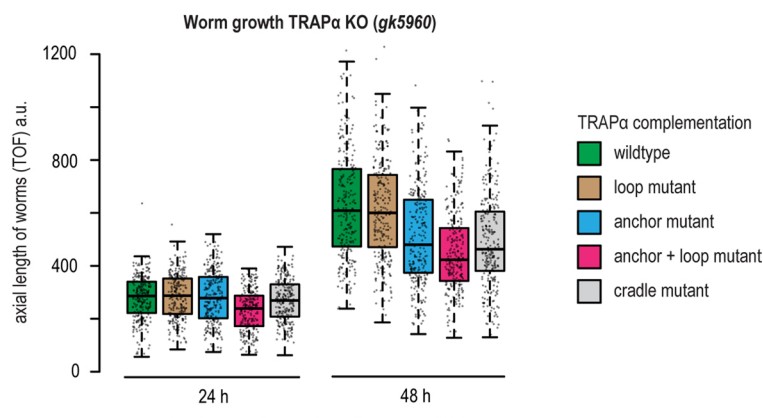

**Extended Data Fig. 10 | Characterization of TRAPα KO complemented worms. a**, FLAG immunoblot analysis showing expression levels of indicated TRAPα variants in C. elegans worms analyzed in main Fig. 3. The different TRAPα variants are tagged with a C-terminal mono-FLAG tag. Gapdh served as loading control. **b**, Growth analysis of TRAPα knockout worms complemented with indicated TRAPα transgenes. Worms were synchronized via a timed egg-lay and plates incubated for the indicated time at 25 °C. The axial length of worms was determined by worm flow cytometry using a COPAS large particle Biosorter. Box plot center line indicates the median, box length the upper and lower quartile, and whiskers the minimum/maximum quartile (n = 253). Dots represent individual data points. TOF, time-of-flight. a.u., arbitrary units.

# Reporting Summary

## Statistics

For all statistical analyses, confirm that the following items are present in the figure legend, table legend, main text, or Methods section.

| n/a | Confirmed | |
|---|---|---|
| ☐ | ☒ | The exact sample size (*n*) for each experimental group/condition, given as a discrete number and unit of measurement |
| ☒ | ☐ | A statement on whether measurements were taken from distinct samples or whether the same sample was measured repeatedly |
| ☒ | ☐ | The statistical test(s) used AND whether they are one- or two-sided *Only common tests should be described solely by name; describe more complex techniques in the Methods section.* |
| ☒ | ☐ | A description of all covariates tested |
| ☒ | ☐ | A description of any assumptions or corrections, such as tests of normality and adjustment for multiple comparisons |
| ☐ | ☒ | A full description of the statistical parameters including central tendency (e.g. means) or other basic estimates (e.g. regression coefficient) AND variation (e.g. standard deviation) or associated estimates of uncertainty (e.g. confidence intervals) |
| ☒ | ☐ | For null hypothesis testing, the test statistic (e.g. *F*, *t*, *r*) with confidence intervals, effect sizes, degrees of freedom and *P* value noted *Give P values as exact values whenever suitable.* |
| ☒ | ☐ | For Bayesian analysis, information on the choice of priors and Markov chain Monte Carlo settings |
| ☒ | ☐ | For hierarchical and complex designs, identification of the appropriate level for tests and full reporting of outcomes |
| ☒ | ☐ | Estimates of effect sizes (e.g. Cohen's *d*, Pearson's *r*), indicating how they were calculated |

*Our web collection on statistics for biologists contains articles on many of the points above.*

## Software and code

Policy information about availability of computer code

| Data collection | EPU 2 (Thermo Fisher) |
|---|---|
| Data analysis | RELION 3.1<br>MotionCorr2<br>Coot 0.9.8.1<br>Phenix 1.20.1<br>UCSF Chimera 1.15<br>UCSF ChimeraX 1.1.1<br>MolProbity<br>AlphaFold2 |

For manuscripts utilizing custom algorithms or software that are central to the research but not yet described in published literature, software must be made available to editors and reviewers. We strongly encourage code deposition in a community repository (e.g. GitHub). See the Nature Portfolio guidelines for submitting code & software for further information.

## Data

Policy information about availability of data

All manuscripts must include a data availability statement. This statement should provide the following information, where applicable:

- Accession codes, unique identifiers, or web links for publicly available datasets
- A description of any restrictions on data availability
- For clinical datasets or third party data, please ensure that the statement adheres to our policy

The data supporting the findings of this study are available in the Electron Microscopy Bank and Protein Data Bank under accession codes EMD-16232 and PDB ID: 8BTK. The structures of rabbit 80S ribosome (PDB 7O7Y) and Sec61 translocon (PDB 6W6L) were used for comparisons and as an initial model. Source data are provided with the manuscript.

## Human research participants

Policy information about studies involving human research participants and Sex and Gender in Research.

| Reporting on sex and gender | Not relevant to this study. |
|---|---|
| Population characteristics | Not relevant to this study. |
| Recruitment | Not relevant to this study. |
| Ethics oversight | Not relevant to this study. |

Note that full information on the approval of the study protocol must also be provided in the manuscript.

# Field-specific reporting

Please select the one below that is the best fit for your research. If you are not sure, read the appropriate sections before making your selection.

☒ Life sciences  ☐ Behavioural & social sciences  ☐ Ecological, evolutionary & environmental sciences

For a reference copy of the document with all sections, see nature.com/documents/nr-reporting-summary-flat.pdf

# Life sciences study design

All studies must disclose on these points even when the disclosure is negative.

| Sample size | No sample size calculations were peformed. The sample size for the worm growth analysis was determined by the number of eggs that were laid during the synchronization step. |
|---|---|
| Data exclusions | No data were excluded from the analyses. |
| Replication | Each worm experiment was independently repeated three times with similar results. |
| Randomization | Samples were not randomized because there is nothing to randomize. The different conditions being compared within any given experiment derive from a single common stock. |
| Blinding | Not relevant to this study. The data were determined by technical means and not by human judgment. |

# Reporting for specific materials, systems and methods

We require information from authors about some types of materials, experimental systems and methods used in many studies. Here, indicate whether each material, system or method listed is relevant to your study. If you are not sure if a list item applies to your research, read the appropriate section before selecting a response.

## Materials & experimental systems

| n/a | Involved in the study |
|---|---|
| ☐ | ☒ Antibodies |
| ☒ | ☐ Eukaryotic cell lines |
| ☒ | ☐ Palaeontology and archaeology |
| ☐ | ☒ Animals and other organisms |
| ☒ | ☐ Clinical data |
| ☒ | ☐ Dual use research of concern |

## Methods

| n/a | Involved in the study |
|---|---|
| ☒ | ☐ ChIP-seq |
| ☒ | ☐ Flow cytometry |
| ☒ | ☐ MRI-based neuroimaging |

## Antibodies

| | |
|---|---|
| Antibodies used | FLAG (Sigma, #F7425), GAPDH (Proteintech, #60004-1-lg), Tubulin (DSHB, clone AA4.3). A table of all antibodies used in this study, their catalog number, and specific conditions for use, are provided in Supplementary Table 1. |
| Validation | Each antibody was validated for specificity against the antigen in immunoblot analyses by the manufacturerer. FLAG (Sigma, #F7425) and GAPDH (Proteintech, #60004-1-lg) antibodies were used in over 100 different publications, and the Tubulin (DSHB, clone AA4.3) antibody in 39 publications, according to Scicrunch.org. |

## Animals and other research organisms

Policy information about studies involving animals; ARRIVE guidelines recommended for reporting animal research, and Sex and Gender in Research

| | |
|---|---|
| Laboratory animals | Caenorhabditis elegans (strain N2, strain SJ4005, strain VC4892, strain DR1572). Animals were studied in the larval and adult day stage. |
| Wild animals | This study did not involve wild animals. |
| Reporting on sex | Not relevant to this study. Only hermaphrodites were analyzed. |
| Field-collected samples | This study did not involve samples collected from the field. |
| Ethics oversight | Not relevant to this study. |

Note that full information on the approval of the study protocol must also be provided in the manuscript.

