## [Peer Review File · Nature Structural & Molecular Biology]

Peer Review Information

Manuscript Title: Molecular basis of TRAP complex function in ER protein biogenesis

Corresponding author name(s): Nenad Ban, Elke Deuerling, Ahmad Jomaa

Reviewer Comments & Decisions:

Decision Letter, initial version:
--

Message: 14th Nov 2022

Dear Dr. Ban,

Thank you again for submitting your manuscript "Molecular basis of the TRAP complex function in ER protein biogenesis". I sincerely apologize for the delay in responding, which resulted from the difficulty in obtaining suitable referee reports, together with our editorial team having been short-staffed in the last months. Nevertheless, we now have comments (below) from the 2 reviewers who evaluated your paper. In light of those reports, we remain interested in your study and would like to see your response to the comments of the referees, in the form of a revised manuscript.

You will see that while the reviewers appreciate the mechanistic insights obtained from the higher resolution structure, Reviewer #1 proposes a reinterpretation of the functional mutagenesis experiments in *C.elegans*, and Reviewer #2 suggests some clarification in the presentation of the structural data. Please be sure to address/respond to all concerns of the referees in full in a point-by-point response and highlight all changes in the revised manuscript text file. If you have comments that are intended for editors only, please include those in a separate cover letter.

We expect to see your revised manuscript within 6 weeks. If you cannot send it within this time, please contact us to discuss an extension; we would still consider your revision, provided that no similar work has been accepted for publication at NSMB or published elsewhere.

As you already know, we put great emphasis on ensuring that the methods and statistics reported in our papers are correct and accurate. As such, if there are any changes that should be reported, please submit an updated version of the Reporting Summary along

with your revision.

Reporting Summary:

Please note that all key data shown in the main figures as cropped gels or blots should be presented in uncropped form, with molecular weight markers. These data can be aggregated into a single supplementary figure item. While these data can be displayed in a relatively informal style, they must refer back to the relevant figures. These data should be submitted with the final revision, as source data, prior to acceptance, but you may want to start putting it together at this point.

Data availability: this journal strongly supports public availability of data. All data used in accepted papers should be available via a public data repository, or alternatively, as Supplementary Information. If data can only be shared on request, please explain why in

your Data Availability Statement, and also in the correspondence with your editor. Please note that for some data types, deposition in a public repository is mandatory - more information on our data deposition policies and available repositories can be found below: <https://www.nature.com/nature-research/editorial-policies/reporting-standards#availability-of-data>

[Redacted]

Sincerely,
Sara

Sara Osman, Ph.D.
Associate Editor
Nature Structural & Molecular Biology

Referee expertise:

Referee #1: *C. elegans* biology, membrane transport

Referee #2: Cryo-EM, protein quality control

Reviewers' Comments:

Reviewer #1:

Remarks to the Author:

This manuscript uses cryo-EM to determine the structure of the Translocon Associated Protein (TRAP) complex that interacts with the Sec translocon and the translating ribosome. The authors identified several features in the complex that may explain how the three complexes interact and coordinately enable the biogenesis of secreted or transmembrane substrate proteins at the endoplasmic reticulum (ER). Phenotypes in *C. elegans* with mutations disrupting these features appear to support their functional importance. Data presented are of good quality and the use of *C. elegans* mutants to define functional importance of identified structural features in a living organismic setting is an applauded approach. However, uncertain novelty as described and flawed interpretation of the functional data dampen the enthusiasm of this reviewer. The authors may consider addressing the following points.

1. Given previous publications (PMID: 18611385, PMID: 28218252, and refs in PMID: 32019826) on structural studies of TRAP/SEC/ribosome complex, novelty of this work remains better defined. It would help readers, especially non-structural biologists outside this field, if the authors can clearly point out what findings are new or different compared to previous studies, and how these new and/or different findings sufficiently advance the field.
2. A major concern lies in the interpretation of the phenotype observed in *C. elegans* mutants, regarding the importance of TRAP for insulin secretion. The authors observed a high rate of dauer formation in *daf-2* mutants, as expected, but the phenotype was suppressed by loss-of-function mutations in TRAP genes. The authors suggested that the suppression was caused by insulin secretion defects, which were not shown. More importantly, insulin secretion defects should mimic, rather than suppress, *daf-2* mutant phenotypes since normal functions of insulin and DAF-2 (insulin receptor) are to inhibit dauer formation. The data appear more consistent with defective secretion of unidentified dauer-promoting factors, rather than insulin itself.
3. Mutations corresponding to the structural features led to ER stress phenotypes, but their specific effects on protein-protein interactions remain unclear. To fully support their conclusions, the authors need consider the possibility whether these mutations, especially those affecting hydrophobic residues, may or may not affect protein abundance, stability or folding.
4. Minor: In describing *C. elegans* reagents and results in Figures, please follow the standard *C. elegans* nomenclature. Gene and species not protein names should be italicized. Multiple alleles should be separated by semicolon not slash. Mutants shown in Figures should carry specific allele information etc.

Reviewer #2:

Remarks to the Author:

This manuscript presents a cryo-EM structure of the translocon associated protein (TRAP) complex bound to a translating ribosome associated with the Sec61 translocon. The improved resolution of the cryo-EM structure compared to previous reports together with AlphaFold-mediated modeling allowed the authors to identify previously unknown interactions between TRAP subunits with the ribosome and with the luminal side of the Sec61 complex. They were further able to demonstrate the physiological importance of these interactions by assaying reporters of ER stress in *C. elegans* strains expressing various TRAP mutants. Altogether, although proposed mechanistic models remain to be fully validated, this study presents new information that furthers our understanding of TRAP interactions.

Specific comments:

1. It is worth comparing these findings to other recent preprints reporting TRAP structures (www.biorxiv.org/content/10.1101/2022.09.28.509949v1, www.biorxiv.org/content/10.1101/2022.09.30.510141v1) and discuss differences, such as the placement of the TRAP α transmembrane domain and interpretations.
2. The description that TRAP "also binds translating ribosomes" (pg 1 line 13) suggests that it was previously shown that TRAP directly contacts ribosomes. However, it is not clear which cited study showed this. From my understanding, it was generally hypothesized that TRAP associated with translating RNCs at the ER via Sec61 rather than through specific interactions with the ribosomes. The authors may want to consider clarifying this point as it may further emphasize the importance of the interactions between TRAP and ribosomes that they observe.
3. Please include map contour levels in the figure legends, including in Fig. 1b,c, Extended Data Fig. 3a-c, Extended Data Fig. 4, Extended Data Fig. 5, and Extended Data Fig. 6.
4. In Fig. 2c, it is not clear why a directly comparable WT control with the same reporters as TRAP KO (myo-2p::GFP and myo-2p::GFP, hsp-4p::GFP) is not shown.
5. It is helpful for reader interpretation to explicitly state the TRAP mutations (e.g. for the anchor and cradle mutants) analyzed in the main text rather than only in Extended Data Table S4.
6. It may be useful to label TM5/6 In Fig. 3b.

Author Rebuttal to Initial comments

Reviewer #1:

Remarks to the Author:

This manuscript uses cryo-EM to determine the structure of the Translocon Associated Protein (TRAP) complex that interacts with the Sec translocon and the translating ribosome. The authors identified several features in the complex that may explain how the three complexes interact and coordinately enable the biogenesis of secreted or transmembrane substrate proteins at the endoplasmic reticulum (ER). Phenotypes in *C. elegans* with mutations disrupting these features appear to support their functional importance. Data presented are of good quality and the use of *C. elegans* mutants to define functional importance of identified structural features in a living organismic setting is an applauded approach. However, uncertain novelty as described and flawed interpretation of the functional data dampen the enthusiasm of this reviewer. The authors may consider addressing the following points.

1. Given previous publications (PMID: 18611385, PMID: 28218252, and refs in PMID: 32019826) on structural studies of TRAP/SEC/ribosome complex, novelty of this work remains better defined. It would help readers, especially non-structural biologists outside this field, if the authors can clearly point out what findings are new or different compared to previous studies, and how these new and/or different findings sufficiently advance the field.

We added a paragraph at the end of the discussion that summarizes our findings, indicating which are new and how they expand our understanding of TRAP participation in the biogenesis and translocation of proteins in the ER.

2. A major concern lies in the interpretation of the phenotype observed in *C. elegans* mutants, regarding the importance of TRAP for insulin secretion. The authors observed a high rate of dauer formation in *daf-2* mutants, as expected, but the phenotype was suppressed by loss-of-function mutations in TRAP genes. The authors suggested that the suppression was caused by insulin secretion defects, which were not shown. More importantly, insulin secretion defects should mimic, rather than suppress, *daf-2* mutant phenotypes since normal functions of insulin and DAF-2 (insulin receptor) are to inhibit dauer formation. The data appear more consistent with defective secretion of unidentified dauer-promoting factors, rather than insulin itself.

We thank the reviewer for this comment and agree with the reviewer that activation of DAF-2/InsR inhibits dauer formation in *C. elegans*. Our interpretation of the DAF-2 data is based on a previous publication in which dauer formation in the DAF-2 mutant was used as an indicator of an insulin secretion defect in *C. elegans* TRAP knockout worms (Li et al., 2019 *Sci. Adv.* PMID: 31840061). In this work, the authors found, as we did, that knockout of TRAP α prevents dauer formation in the *daf-2(e1368)* mutant. These authors further show that TRAP α knockout only suppresses the dauer-constitutive phenotype of mutants with reduced DAF-2/InsR signaling, but not of mutants with reduced signaling in other pathways that affect dauer formation. The authors therefore concluded that TRAP promotes dauer arrest by specifically antagonizing the DAF-2/InsR pathway. *C. elegans* expresses 40 different insulin-like peptides, some of which enhance dauer arrest by antagonizing DAF-2/InsR signaling (PMID: 11274053, PMID: 24671950). For example, INS-1, the closest relative of human insulin, is a DAF-2/InsR antagonist that promotes dauer formation in *C. elegans* (PMID: 11274053). Similarly, human insulin expressed in *C. elegans* antagonizes DAF-2/InsR signaling and promotes dauer formation (PMID: 11274053). Because insulin secretion in human cells depends on TRAP (PMID: 31840061, PMID: 33137310), Li et al. concluded that the suppression of dauer formation in TRAP α knockout worms was most likely due to a secretion defect of insulin-like peptides that antagonize DAF-2/InsR.

In the revised manuscript, we modified the text to better explain the interpretation of the DAF-2 data clarifying that secretion of insulin-like peptides that antagonize DAF-2/InsR is most likely affected by

the TRAP mutants based on the findings by Li et al (PMID: 31840061). We also agree with the reviewer that there are other possibilities that could explain the suppression of dauer formation in DAF-2/InsR mutants by TRAP knockout and mention this in the text.

3. Mutations corresponding to the structural features led to ER stress phenotypes, but their specific effects on protein-protein interactions remain unclear. To fully support their conclusions, the authors need consider the possibility whether these mutations, especially those affecting hydrophobic residues, may or may not affect protein abundance, stability or folding.

The expression levels of all TRAP α and TRAP γ variants were analyzed by immunoblotting through detection of the C-terminal FLAG tag. All variants were robustly expressed at comparable levels to the wildtype variant, suggesting all mutants form a stable TRAP complex. Data are shown in Supplementary Figure 8a-c and Supplementary Figure 9a.

4. Minor: In describing *C. elegans* reagents and results in Figures, please follow the standard *C. elegans* nomenclature. Gene and species not protein names should be italicized. Multiple alleles should be separated by semicolon not slash. Mutants shown in Figures should carry specific allele information etc.

We agree with the reviewer and changed the figure descriptions to standard *C. elegans* nomenclature.

Reviewer #2:

Remarks to the Author:

This manuscript presents a cryo-EM structure of the translocon associated protein (TRAP) complex bound to a translating ribosome associated with the Sec61 translocon. The improved resolution of the cryo-EM structure compared to previous reports together with AlphaFold-mediated modeling allowed the authors to identify previously unknown interactions between TRAP subunits with the ribosome and with the luminal side of the Sec61 complex. They were further able to demonstrate the physiological importance of these interactions by assaying reporters of ER stress in *C. elegans* strains expressing various TRAP mutants. Altogether, although proposed mechanistic models remain to be fully validated, this study presents new information that furthers our understanding of TRAP interactions.

Specific comments:

1. It is worth comparing these findings to other recent preprints reporting TRAP structures (www.biorxiv.org/content/10.1101/2022.09.28.509949v1, www.biorxiv.org/content/10.1101/2022.09.30.510141v1) and discuss differences, such as the placement of the TRAP α transmembrane domain and interpretations.

According to the reviewer suggestion we added a short statement to the discussion section, in which we mentioned the recent preprints. Additionally, we indicated that none of the two structures describe the specific TRAP α anchor interaction with the ribosome that was discovered and investigated *in vivo* in this study.

However, we chose not to discuss other differences, such as the different placement of the TRAP α transmembrane domain in Pauwels et al., as some of these interpretations and models may change during peer-review process. Although, such comparisons are difficult without the described cryo-EM

maps and/or molecular models, it appears that the position of TRAP α anchor is not consistent with the placement of the TRAP α transmembrane domain in the helical bundle with other TRAP subunits as described by Pauwels et al.

2. The description that TRAP “also binds translating ribosomes” (pg 1 line 13) suggests that it was previously shown that TRAP directly contacts ribosomes. However, it is not clear which cited study showed this. From my understanding, it was generally hypothesized that TRAP associated with translating RNCs at the ER via Sec61 rather than through specific interactions with the ribosomes. The authors may want to consider clarifying this point as it may further emphasize the importance of the interactions between TRAP and ribosomes that they observe.

A reference to a study by Pfeffer et al. (PMID: 32019826) was added to the sentence mentioned by the reviewer. The cited study indeed showed that the TRAP complex interacts with a ribosome via a cytoplasmic domain of TRAP γ . However, without an atomic model, neither the nature of these interactions nor the residues involved have been suggested. Our structural analysis, combined with the *in vivo* experiments allowed us to characterize these interactions in more detail.

3. Please include map contour levels in the figure legends, including in Fig. 1b,c, Extended Data Fig. 3a-c, Extended Data Fig. 4, Extended Data Fig. 5, and Extended Data Fig. 6.

Sigma values have been added to the figure legends as suggested.

4. In Fig. 2c, it is not clear why a directly comparable WT control with the same reporters as TRAP KO (myo-2p::GFP and myo-2p::GFP, hsp-4p::GFP) is not shown.

The pharynx marker (myo-2p::GFP) is only present in the TRAP α KO strain (the TRAP α gene is replaced by myo2p::GFP). Thus, we cannot cross only the myo-2p::GFP cassette with hsp-4p::GFP because the TRAP α -KO allele would co-segregate. However, we show the TRAP α KO+myo-2p::GFP/hsp-4p::GFP strain complemented with a wildtype TRAP α gene in Fig. 2d, Fig. 3d and Fig. 3f. As expected, high GFP expression in this strain is observed only in the pharynx.

5. It is helpful for reader interpretation to explicitly state the TRAP mutations (e.g. for the anchor and cradle mutants) analyzed in the main text rather than only in Extended Data Table S4.

We agree with the reviewer and added this information to the main text.

6. It may be useful to label TM5/6 in Fig. 3b.

We added a label indicating the loop between TMH 5/6 of Sec61 in Fig. 3b.

Decision Letter, first revision:**Message:** 5th Jan 2023

Dear Dr. Ban,

Thank you again for submitting your manuscript "Molecular basis of the TRAP complex function in ER protein biogenesis". I apologize for the delay in responding, which resulted from the difficulty in obtaining suitable referee reports together with a significantly reduced editorial capacity over the holidays. Nevertheless, we now have comments (below) from the 2 reviewers who evaluated your paper. In light of those reports, we remain interested in your study and would like to see your response to the comments of the referees, in the form of a revised manuscript.

You will see that there are still a few outstanding concerns regarding the lack of evidence to support insulin secretion as the functional mechanism, which requires further toning down, as well as a remaining concern about the structural modeling of certain regions. Please be sure to address/respond to all concerns of the referees in full in a point-by-point response and highlight all changes in the revised manuscript text file. If you have comments that are intended for editors only, please include those in a separate cover letter.

We expect to see your revised manuscript within 6 weeks. If you cannot send it within this time, please contact us to discuss an extension; we would still consider your revision, provided that no similar work has been accepted for publication at NSMB or published elsewhere.

Reporting Summary:

When submitting the revised version of your manuscript, please pay close attention to our [href="https://www.nature.com/nature-portfolio/editorial-policies/image-integrity">Digital Image Integrity Guidelines. and to the following points below:](https://www.nature.com/nature-portfolio/editorial-policies/image-integrity)

- that unprocessed scans are clearly labelled and match the gels and western blots presented in figures.
- that control panels for gels and western blots are appropriately described as loading on

sample processing controls

-- all images in the paper are checked for duplication of panels and for splicing of gel lanes.

Please note that all key data shown in the main figures as cropped gels or blots should be presented in uncropped form, with molecular weight markers. These data can be aggregated into a single supplementary figure item. While these data can be displayed in a relatively informal style, they must refer back to the relevant figures. These data should be submitted with the final revision, as source data, prior to acceptance, but you may want to start putting it together at this point.

Data availability: this journal strongly supports public availability of data. All data used in accepted papers should be available via a public data repository, or alternatively, as Supplementary Information. If data can only be shared on request, please explain why in your Data Availability Statement, and also in the correspondence with your editor. Please note that for some data types, deposition in a public repository is mandatory - more information on our data deposition policies and available repositories can be found below: <https://www.nature.com/nature-research/editorial-policies/reporting-standards#availability-of-data>

While we encourage the use of color in preparing figures, please note that this will incur a charge to partially defray the cost of printing. Information about color charges can be

found at <http://www.nature.com/nsmb/authors/submit/index.html#costs>

[Redacted]

Sincerely,
Sara

Sara Osman, Ph.D.
Associate Editor
Nature Structural & Molecular Biology

Reviewers' Comments:

Reviewer #1:

Remarks to the Author:

The authors have improved the manuscript by addressing the reviewers' comments, but still not sufficiently addressed the concern on "insulin secretion". The claims on insulin secretion were made multiple times in Abstract and main text, yet there is no experimental evidence throughout the manuscript that points to "insulin secretion" as the underlying cause of rescue by loss of TRAP in *daf-2* mutants. The clear epistasis relationship between TRAP and *daf-2* mutations suggests TRAP functions normally downstream or in parallel to DAF-2, not the other way around. As dauer formation in *daf-2* mutants require the transcription factor DAF-16, the authors' result can be explained by an unidentified DAF-16 target gene that encodes a dauer-promoting factor and requires TRAP for secretion. Such factor can be insulin or independent of insulin. I am not convinced by the claim Unless the authors provide direct evidence for such insulin secretion being causal. If structural insights of TRAP are sufficiently novel and important, the authors might consider simply tone down "insulin secretion".

Reviewer #2:

Remarks to the Author:

The revised manuscript incorporates several changes in response to the first round of reviews. However, I have two remaining sticking points:

1. The placement of the TRAP α TMH. I agree with the authors' suggestion that interpretations regarding the placement of the TRAP α TMH in the other preprints may change. However, related to this point, Extended Data Fig. 5a is not very convincing regarding its placement in these models. Is there a clearer view of the helix density? Would it be more appropriate to leave it unmodeled? Alternatively, please provide the maps and models that led to this interpretation.

2. A minor but general weakness of the study is the relatively large gap between the molecular details identified from the structural analysis and the ER stress phenotypes analyzed in *C. elegans*. Although the functional analyses are beautifully done, the assays only indicate that the conserved residues analyzed are important (which may not be surprising because they're conserved) and do not directly address the mechanistic interpretations, such as when TRAP engages ribosomes during ER targeting or nascent protein chaperoning by the hydrophobic cradle. In the absence of assays specifically interactions between TRAP and the ribosome, nascent protein, or Sec61 (which would be beyond the scope of revisions) or controls in which conserved residues not identified to be important based on the structural models are mutated, it would be useful to explicitly point out this limit on mechanistic interpretations. This is especially true for the extensive cradle mutant in Fig. 3. It seems conceivable, for example, that these mutations may also disrupt interaction with Sec61.

Author Rebuttal, first revision:

Reviewer #1:

Remarks to the Author:

The authors have improved the manuscript by addressing the reviewers' comments, but still not sufficiently addressed the concern on "insulin secretion". The claims on insulin secretion were made multiple times in Abstract and main text, yet there is no experimental evidence throughout the manuscript that points to "insulin secretion" as the underlying cause of rescue by loss of TRAP in daf-2 mutants. The clear epistasis relationship between TRAP and daf-2 mutations suggests TRAP functions normally downstream or in parallel to DAF-2, not the other way around. As dauer formation in daf-2 mutants require the transcription factor DAF-16, the authors' result can be explained by an unidentified DAF-16 target gene that encodes a dauer-promoting factor and requires TRAP for secretion. Such factor can be insulin or independent of insulin. I am not convinced by the claim Unless the authors provide direct evidence for such insulin secretion being causal. If structural insights of TRAP are sufficiently novel and important, the authors might consider simply tone down "insulin secretion".

We thank the reviewer for this comment. Although, the role of TRAP for insulin secretion has been established by direct evidence in human cells by two different groups (Li et al., Sci Adv. 2019; Kriegler et al., J Mol Biol. 2020), we agree with the reviewer that neither our study nor that of Li et al. directly assess insulin levels in the *C. elegans* model system. Therefore, we cannot exclude the possibility that defective secretion of a factor other than insulin accounts for the observed phenotype of the insulin receptor mutant. This is stated in the RESULTS section of the manuscript: "*Defective secretion of other dauer-promoting factors unrelated to insulin could also contribute to the suppression of dauer formation in these mutants*". As suggested by the reviewer, we further toned down the "insulin secretion" argument in the ABSTRACT by changing the text to "protein hormone secretion". Nevertheless, we consider it reasonable to mention the requirement of TRAP for insulin production when referring to previously published papers.

Reviewer #2

Remarks to the Author:

The revised manuscript incorporates several changes in response to the first round of reviews. However, I have two remaining sticking points:

1. The placement of the TRAP α TMH. I agree with the authors' suggestion that interpretations regarding the placement of the TRAP α TMH in the other preprints may change. However, related to this point, Extended Data Fig. 5a is not very convincing regarding its placement in these models. Is there a clearer view of the helix density? Would it be more appropriate to leave it unmodeled? Alternatively, please provide the maps and models that led to this interpretation.

We thank the reviewer for this comment. We have taken multiple considerations into account when interpreting the density that we attributed to TRAP α TMH however, this was not adequately described in the manuscript.

To better explain our modeling, that included the AlphaFold prediction of the TRAP α as a self-standing TMH, the observation of a single transmembrane helix in the density and the connectivity considerations, we introduced the following statement in the main text:

The location of the C-terminus anchor of the TRAP α subunit on the surface of the ribosome “also supports the placement of the free-standing TRAP α TMH, as a linker bridging these two elements is too short to reach the density corresponding to the other TRAP TMHs (Extended Data Fig. 6d,e). Furthermore, such placement of the TRAP α TMH positions a conserved region of positively charged residues next to the 5.8S rRNA (Extended Data Fig. 6f,g), allowing for favorable electrostatic interactions.”

As shown below, the Extended Data Figure 6 mentioned in this paragraph shows additional cryo-EM map views (Extended Data Fig. 6a-c) and other structural aspects that we considered during model building (Extended Data Fig. d-g).

2. A minor but general weakness of the study is the relatively large gap between the molecular details identified from the structural analysis and the ER stress phenotypes analyzed in *C. elegans*. Although the functional analyses are beautifully done, the assays only indicate that the conserved residues analyzed are important (which may not be surprising because they're conserved) and do not directly address the mechanistic interpretations, such as when TRAP engages ribosomes during ER targeting or nascent protein chaperoning by the hydrophobic cradle. In the absence of assays specifically interactions between TRAP and the ribosome, nascent protein, or Sec61 (which would be beyond the scope of revisions) or controls in which conserved residues not identified to be important based on the structural models are mutated, it would be useful to explicitly point out this limit on mechanistic interpretations. This is especially true for the extensive cradle mutant in Fig. 3. It seems conceivable, for example, that these mutations may also disrupt interaction with Sec61.

We thank the reviewer for their comment and suggestions. Indeed, our study provide the structural and functional basis of TRAP and underscores its role in ribosome and Sec translocon interaction to facilitate membrane protein biogenesis. Purifying TRAP complex and performing mutational studies will be imperative for future investigations to further understand mechanism underlying TRAP function, however, as also pointed by the reviewer, this beyond the scope of the current study.

We introduced a statement at the end of discussion mentioning the importance of future experiments in studying the function of TRAP in membrane protein biogenesis: *"Nevertheless, to fully understand its mechanism of action and substrate specificity in membrane protein biogenesis further experiments, including mutational studies with purified TRAP complex, will be critical."*

Decision Letter, second revision:

Message: Our ref: NSMB-A46757B

9th Feb 2023

Dear Dr. Ban,

Thank you for submitting your revised manuscript "Molecular basis of the TRAP complex function in ER protein biogenesis" (NSMB-A46757B). It has now been assessed by the editorial team. We find that the paper has improved in revision, and therefore we'll be happy in principle to publish it in Nature Structural & Molecular Biology, pending minor revisions to comply with our editorial and formatting guidelines, and to satisfy outstanding referee requests, should there be any remaining unaddressed.

To facilitate our work at this stage, we would appreciate if you could send us the main text as a word file. Please make sure to copy the NSMB account (cc'ed above).

Sincerely,
Sara

Sara Osman, Ph.D.
Associate Editor
Nature Structural & Molecular Biology

Final Decision Letter:

Message 6th Apr 2023

:

Dear Dr. Ban,

We are now happy to accept your revised paper "Molecular basis of the TRAP complex function in ER protein biogenesis" for publication as a Article in Nature Structural & Molecular Biology.

Your paper will be published online soon after we receive proof corrections and will appear in print in the next available issue. You can find out your date of online publication by contacting the production team shortly after sending your proof corrections. Content is published online weekly on Mondays and Thursdays, and the embargo is set at 16:00 London time (GMT)/11:00 am US Eastern time (EST) on the day of publication. Now is the time to inform your Public Relations or Press Office about your paper, as they might be interested in promoting its publication. This will allow them time to prepare an accurate and satisfactory press release. Include your manuscript tracking number (NSMB-A46757C) and our journal name, which they will need when they contact our press office.

About one week before your paper is published online, we shall be distributing a press release to news organizations worldwide, which may very well include details of your work. We are happy for your institution or funding agency to prepare its own press release, but it must mention the embargo date and Nature Structural & Molecular Biology. If you or your Press Office have any enquiries in the meantime, please contact press@nature.com.

You can now use a single sign-on for all your accounts, view the status of all your

manuscript submissions and reviews, access usage statistics for your published articles and download a record of your refereeing activity for the Nature journals.

Please note that *Nature Structural & Molecular Biology* is a Transformative Journal (TJ). Authors may publish their research with us through the traditional subscription access route or make their paper immediately open access through payment of an article-processing charge (APC). Authors will not be required to make a final decision about access to their article until it has been accepted. <https://www.springernature.com/gp/open-research/transformative-journals> Find out more about Transformative Journals

Sincerely,
Sara

Sara Osman, Ph.D.
Associate Editor
Nature Structural & Molecular Biology
